# A Functional Characterization of Randomly Initialized Gradient Descent in Deep ReLU Networks

## Abstract

Despite their popularity and successes, deep neural networks are poorly understood theoretically and treated as 'black box' systems. Using a functional view of these networks gives us a useful new lens with which to understand them. Theoretical (in shallow) and experimentally probing properties of these networks reveals insights into the effect of standard initializations, the value of depth, the underlying loss surface, and the origins of generalization. One key result is that generalization results from smoothness of the functional approximation, combined with a flat initial approximation. This smoothness increases with number of units, explaining why massively overparamaterized networks continue to generalize well.

## 1 Introduction

Deep neural networks, trained via gradient descent, have revolutionized the field of machine learning. Despite their widespread adoption, theoretical understanding of fundamental properties of deep learning – the true value of depth, the root cause of implicit regularization, and the seemingly 'unreasonable' generalization achieved by overparameterized networks – remains mysterious.

Empirically, it is known that depth is critical to the success of deep learning. Theoretically, it has been proven that maximum expressivity grows exponentially with depth, with a smaller number of trainable parameters (Raghu et al., 2017; Poole et al., 2016). This theoretical capacity may not be used, as recently shown explicitly by (Hanin & Rolnick, 2019). Instead, the number of regions within a trained network is proportional to the total number of hidden units, regardless of depth. Clearly deep networks perform better, but what is the value of depth if not in increasing expressivity?

Another major factor leading to the success and widespread adoption of deep learning has been its surprisingly high generalization performance (Zhang et al., 2016). In contrast to other machine learning techniques, continuing to add parameters to a deep network (beyond zero training loss) tends to *improve* generalization performance. This is even for networks that are massively overparameterized, wherein according to traditional ML theory they should (over)fit all the training data (Neyshabur et al., 2015). How does training deep networks with excess capacity lead to generalization? And how can it be that this generalization error decreases with overparameterization?

We believe that taking a functional view allows us a new, useful lens with which to explore and understand these issues. In particular, we focus on shallow and deep fully connected univariate ReLU networks, whose parameters will always result in a Continuous Piecewise Linear (CPWL) approximation to the target function. We provide theoretical results for shallow networks, with experiments showing that these qualitative results hold in deeper nets.

Our approach is related to previous work from (Savarese et al., 2019; Arora et al., 2019; Frankle & Carbin, 2018) in that we wish to characterize parameterization and generalization. We differ from these other works by using small widths, rather than massively overparamaterized or infinite, and by using a functional parameterization to measure properties such as smoothness. Other prior works such as (Serra et al., 2017; Arora et al., 2016; Montufar et al., 2014) attempt to provide theoretical upper or lower bounds to the number of induced pieces in ReLU networks, whereas we are more interested in the empirical number of pieces in example tasks. Interestingly, (Serra et al., 2017) also takes a functional view, but is not interested in training and generalization as we are. Previous work

(Advani & Saxe, 2017) has hinted at the importance of small norm initialization, but the functional perspective allows us to prove generalization properties in shallow networks.

**Main Contributions** The main contribution of this work are as follows:

- *Functional Perspective of Initialization: Increasingly Flat with Depth.* In the functional perspective, neural network parameters determine the locations of breakpoints and their delta-slopes (defined in Section 2.1) in the CPWL reparameterization. We prove that, for common initializations, these distributions are mean 0 with low standard deviation. The delta-slope distribution becomes increasingly concentrated as the depth of the network increases, leading to flatter approximations. In contrast, the breakpoint distribution grows wider, allowing deeper network to better approximate over a broader range of inputs.

- *Value of Depth: Optimization, not Expressivity.* Theoretically, depth adds an exponential amount of expressivity. Empirically, this is not true in trained deep networks. We find that expressivity scales with the number of total units, and weakly if at all with depth. However, we find that depth makes it easier for GD to optimize the resulting network, allowing for a greater flexibility in the movement of breakpoints, as well as the number of breakpoints induced during training.

- *Generalization is due to Flat Initialization in the Overparameterized Regime.* We find that generalization in overparametrized FC ReLu nets is due to three factors: (i) the very flat initialization, (ii) the curvature-based parametrization of the approximating function (breakpoints and delta-slopes) and (iii) the role of gradient descent (GD) in preserving (i) and regularizing via (ii). In particular, the global, rather than local, impact of breakpoints and delta-slopes helps regularize the approximating function in the large gaps between training data, resulting in their smoothness. Due to these nonlocal effects, more overparameterization leads to smoother approximations (all else equal), and thus typically better generalization (Neyshabur et al., 2018; 2015).

## 2 THEORETICAL RESULTS

### 2.1 RELU NETS IN FUNCTION SPACE: FROM WEIGHTS TO BREAKPOINTS & SLOPES

Consider a fully connected ReLU neural net $\hat{f}_\theta(x)$ with a single hidden layer of width $H$, scalar input $x \in \mathbb{R}$ and scalar output $y \in \mathbb{R}$. $\hat{f}(\cdot; \theta)$ is continuous piecewise linear function (CPWL) since the ReLU nonlinearity is CPWL. We want to understand the *function* implemented by this neural net, and so we ask: How do the CPWL parameters relate to the NN parameters? We answer this by transforming from the NN parametrization (weights and biases) to two CPWL parametrizations:

$$\hat{f}(x; \theta_{NN}) \triangleq \sum_{i=1}^{H} v_i (w_i x + b_i)_+ \tag{1}$$

$$= \sum_{i=1}^{H} \mu_i (x - \beta_i) \begin{cases} [\![x > \beta_i]\!], & s_i = 1 \\ [\![x < \beta_i]\!], & s_i = -1 \end{cases} \triangleq \hat{f}(x; \theta_{BDSO}) \tag{2}$$

$$= \sum_{p=1}^{P} [\![\beta_p \leq x < \beta_{p+1}]\!] (m_p x + \gamma_p) \triangleq \hat{f}(x; \theta_{PWL}) \tag{3}$$

where the Iversen bracket $[\![b]\!]$ is 1 when the condition $b$ is true, and 0 otherwise. Here the NN parameters $\theta_{NN} \triangleq \{(w_i, b_i, v_i)\}_{i=1}^{H}$ denote the input weight, bias, and output weight of neuron $i$, and $(\cdot)_+ \triangleq \max\{0, \cdot\}$ denotes the ReLU function. The first CPWL parametrization is $\theta_{BDSO} \triangleq \{(\beta_i, \mu_i, s_i)\}_{i=1}^{H}$, where $\beta_i \triangleq -\frac{b_i}{w_i}$ is (the x-coordinate of) the *breakpoint* (or *knot*) induced by neuron $i$, $\mu_i \triangleq w_i v_i$ is the *delta-slope* contribution of neuron $i$, and $s_i \triangleq \operatorname{sgn} w_i \in \{\pm 1\}$ is the *orientation* of $\beta_i$ (left for $s_i = -1$, right for $s_i = +1$). Intuitively, in a good fit the breakpoints $\beta_i$ will congregate in areas of high curvature in the ground truth function $|f''(x)| \geq 0$, while delta-slopes $\mu_i$ will actually implement the needed curvature by changing the slope by $\mu_i$ from one piece $p(i)$ to the next $p(i) + 1$. As the number of pieces grows, the approximation will improve, and the delta-slopes (scaled by the piece lengths) approach the true curvature of $f$: $\lim_{P \to \infty} \mu_{p(i)}/(\beta_p - \beta_{p-1}) \to f''(x = \beta_i)$.

We note that the BDSO parametrization of a ReLU NN is closely related to but different than a traditional roughness-minimizing $m$-th order spline parametrization $\hat{f}_{\text{spline}}(x) \triangleq \sum_{i=1}^{K} \mu_i (x - \beta_i)_+^m + \sum_{j=0}^{m} c_j x^j$: BDSO (i) lacks the base polynomial, and (ii) it has two possible breakpoint orientations $s_i \in \{\pm 1\}$ whereas the spline only has one. We note in passing that adding in the base polynomial (for linear case $m = 1$) into the BDSO ReLU parametrization yields a ReLU ResNet parametrization. We believe this is a novel viewpoint that may shed more light on the origin of the effectiveness of ResNets, but we leave it for future work.

The second parametrization is the canonical one for PWL functions: $\theta_{PWL} \triangleq \{(\beta_p, m_p, \gamma_p)\}_{p=1}^{P}$, where $\beta_0 < \beta_1 < \ldots < \beta_p \triangleq -\frac{b_{p(i)}}{w_{p(i)}} < \ldots < \beta_P$ is the sorted list of (the $x$-coordinates of) the $P \triangleq H + 1$ breakpoints (or knots), $m_p, \gamma_p$ are the slope and y-intercept of piece $p$.

Computing the analogous reparametrization to function space for deep networks is more involved, so we present a basic overview here, and a more detailed treatment in Appendix B. For $L \geq 2$ layers with widths $H^{(\ell)}$, the neural network's activations are defined as: $z_i^{(\ell)} = \sum_{j=1}^{H^{(\ell-1)}} w_{ij}^{(\ell)} x_j^{(\ell-1)} + b_i^{(\ell)}, x_i^{(\ell)} = (z_i^{(\ell)})_+, g_\theta(x) = z^{(L+1)}$ for all hidden layers $\ell \in \{1, 2, \ldots, L\}$ and for all neurons $i \in \{1, 2, \ldots, H^{(\ell)}\}$. Then $\beta_i^{(\ell)}$ is a *breakpoint induced by neuron $i$ in layer $\ell$* if it is a zero-crossing of the net input i.e. $z_i^{(\ell)}(\beta_i^{(\ell)}) = 0$.

Considering these parameterizations (especially the BDSO parameterization) provides a new, useful lens with which to analyze neural nets, enabling us to reason more easily and transparently about the initialization, loss surface, and training dynamics. The benefits of this approach derive from two main properties: (1) that we have 'modded out' the degeneracies in the NN parameterization and (2) the loss depends on the NN parameters $\theta_{NN}$ only through the BDSO parameters (the approximating function) $\theta_{BDSO}$ i.e. $\ell(\theta_{NN}) = \ell(\theta_{BDSO}(\theta_{NN}))$, analogous to the concept of a minimum sufficient statistic in exponential family models. Much recent related work has also veered in this direction, analyzing function space (Hanin & Rolnick, 2019; Balestriero et al., 2018).

## 2.2 RANDOM INITIALIZATION IN FUNCTION SPACE

We now study the random initializations commonly used in deep learning in function space. These include the independent Gaussian initialization, with $b_i \sim N(0, \sigma_b)$, $w_i \sim N(0, \sigma_w)$, $v_i \sim N(0, \sigma_v)$, and independent Uniform initialization, with $b_i \sim U[-a_b, a_b]$, $w_i \sim U[-a_w, a_w]$, $v_i \sim U[-a_v, a_v]$. We find that common initializations result in flat functions, becoming flatter with increasing depth.

**Theorem 1.** *Consider a fully connected ReLU neural net with scalar input and output, and a single hidden layer of width $H$. Let the weights and biases be initialized randomly according to a zero-mean Gaussian or Uniform distribution. Then the induced distributions of the function space parameters (breakpoints $\beta$, delta-slopes $\mu$) are as follows:*

*(a) Under an independent Gaussian initialization,*

$$p_{\beta,\mu}(\beta_i, \mu_i) = \frac{1}{2\pi\sigma_v \sqrt{\sigma_b^2 + \sigma_w^2 \beta_i^2}} \exp\left[-\frac{|\mu_i|\sqrt{\sigma_b^2 + \sigma_w^2 \beta_i^2}}{\sigma_b \sigma_v \sigma_w}\right]$$

*(b) Under an independent Uniform initialization,*

$$p_{\beta,\mu}(\beta_i, \mu_i) = \frac{[\![|\mu_i| \leq \min\{\frac{a_b a_v}{|\beta_i|}, a_w, a_v\}]\!]}{4a_b a_w a_v} \left(\min\{\frac{a_b}{|\beta_i|}, a_w\} - \frac{|\mu_i|}{a_v}\right)$$

Using this result, we can immediately derive marginal and conditional distributions for the breakpoints and delta-slopes.

**Corollary 1.** *Consider the same setting as Theorem 1.*

*(a) In the case of an independent Gaussian initialization,*

$$p_\beta(\beta_i) = \text{Cauchy}\left(\beta_i; 0, \frac{\sigma_b}{\sigma_w}\right) = \frac{\sigma_b \sigma_w}{\pi(\sigma_w^2 \beta_i^2 + \sigma_b^2)}$$

$$p_\mu(\mu_i) = \frac{1}{2\pi\sigma_v\sigma_w} G_{0,2}^{2,0}\left(\frac{\mu_i^2}{4\sigma_v\sigma_w}\bigg|0,0\right) = \frac{1}{\pi\sigma_v\sigma_w} K_0\left(\frac{|\mu_i|}{\sigma_v\sigma_w}\right)$$

$$p_{\mu|\beta}(\mu_i|\beta_i) = Laplace\left(\mu_i; 0, \frac{\sigma_b\sigma_v\sigma_w}{\sqrt{\sigma_b^2 + \sigma_w^2\beta_i^2}}\right) = \frac{\sqrt{\sigma_b^2 + \sigma_w^2\beta_i^2}}{2\sigma_b\sigma_v\sigma_w}\exp\left[-\frac{|\mu_i|\sqrt{\sigma_b^2 + \sigma_w^2\beta_i^2}}{\sigma_b\sigma_v\sigma_w}\right],$$

*where $G_{pq}^{nm}(\cdot|\cdot)$ is the Meijer G-function and $K_\nu(\cdot)$ is the modified Bessel function of the second kind.*

*(b)  In the case of an independent Uniform initialization,*

$$p_\beta(\beta_i) = \frac{1}{4a_b a_w}\left(\min\left\{\frac{a_b}{|\beta_i|}, a_w\right\}\right)^2$$

$$p_\mu(\mu_i) = \frac{[\![-a_w a_v \le \mu_i \le a_w a_v]\!]}{2a_w a_v}\log\frac{a_w a_v}{|\mu_i|}$$

$$p_{\mu|\beta}(\mu_i|\beta_i) = Tri(\mu_i; a_v\min\{a_b/|\beta_i|, a_w\}) = \frac{[\![|\mu_i| \le a_v\min\{a_b/|\beta_i|, a_w\}]\!]}{a_v\min\{a_b/|\beta_i|, a_w\}}\left(1 - \frac{|\mu_i|}{a_v\min\{a_b/|\beta_i|, a_w\}}\right),$$

*where $Tri(\cdot; a)$ is the symmetric triangular distribution with base $[-a, a]$ and mode $0$.*

**Implications.** Corollary 1 implies that the breakpoint density drops quickly away from the origin for common initializations. If $f$ has significant curvature far from the origin, then it may be far more difficult to fit. We show that this is indeed the case by training a shallow ReLU NN with an initialization that does not match the underlying curvature, with training becoming easier if the initial breakpoint distribution better matches the function curvature. We also show that during training, breakpoint distributions move to better match the underlying function curvature, and that this effect increases with depth (see Section 3, Table 1, and Appendix A.6). This implies that a data-dependent initialization, with a breakpoint distribution near areas of high curvature, could potentially be faster and easier to train.

Next, we consider the typical Gaussian He (He et al., 2015) or Glorot (Glorot & Bengio) initializations. In the He initialization, we have $\sigma_w = \sqrt{2}$, $\sigma_v = \sqrt{2/H}$. In the Glorot initalization, we have $\sigma_w = \sigma_v = \sqrt{2/(H+1)}$. We wish to consider their effect on the smoothness of the initial function approximation. From here on, we measure the smoothness using a roughness metric, defined as $\rho \triangleq \sum_i \mu_i^2$, where lower roughness indicates a smoother approximation.

**Theorem 2.** *Consider the initial roughness $\rho_0$ under a Gaussian initialization. In the He initialization, we have that the tail probability is given by*

$$\mathbb{P}[\rho_0 - \mathbb{E}[\rho_0] \ge \lambda] \le \frac{1}{1 + \frac{\lambda^2 H}{128}},$$

*where $\mathbb{E}[\rho_0] = 4$. In the Glorot initialization, we have that the tail probability is given by*

$$\mathbb{P}[\rho_0 - \mathbb{E}[\rho_0] \ge \lambda] \le \frac{1}{1 + \frac{\lambda^2(H+1)^4}{128H}},$$

*where $\mathbb{E}[\rho_0] = \frac{4H}{(H+1)^2} = O\left(\frac{1}{H}\right)$.*

Thus, as the width $H$ increases, the distribution of the roughness of the initial function $\hat{f}_0$ gets tighter around its mean. In the case of the He initialization, this mean is constant; in the Glorot initialization, it decreases with $H$. In either case, for reasonable widths, the initial roughness is small with high probability. This smoothness has implications for the implicit regularization/generalization phenomenon observed in recent work (Neyshabur et al., 2018) (see Section 3 for generalization/smoothness analysis during training).

*Related Work.* Several recent works analyze the random initialization in deep networks. However, there are two main differences, First, they focus on the infinite width case (Savarese et al., 2019; Jacot et al., 2018; Lee et al., 2017) and can thus use the Central Limit Theorem (CLT), whereas we focus on finite width case and cannot use the CLT, thus requiring nontrivial mathematical machinery (see Supplement for detailed proofs). Second, they focus on the activations as a function of input whereas we also compute the joint densities of the BDSO parameters i.e. breakpoints and delta-slopes. The latter is particularly important for understanding the non-uniform density of breakpoints away from the origin as noted above.

## 2.3 Loss Surface in the Function Space

We now consider the mean squared error (MSE) loss as a function of either the NN parameters $\ell(\theta_{NN}) \triangleq \sum_{n=1}^{N} \frac{1}{2}(\hat{f}(x_n; \theta) - y_n)^2$ or the BDSO parameters $\tilde{\ell}(\theta_{BDSO})$. Now consider some $\theta_{BDSO}$. Then $\hat{f}(\cdot; \theta_{BDSO})$ induces a partition $\Pi = (\pi_1, \ldots, \pi_{H+1})$ of the data $\{x_n\}_{n=1}^{N}$ such that the restriction of $\hat{f}_{BDSO}$ to any piece of this partition, denoted $\hat{f}(\cdot; \theta_{BDSO})|_{\pi_p}$, is a linear function.

**Theorem 3.** *Suppose that for all $p \in [P]$, $\hat{f}(\cdot; \theta_{BDSO})|_{\pi_p}$ is an Ordinary Least Squares fit of the data in piece $p$. Then, $\theta_{BDSO}$ is a critical point of $\tilde{\ell}(\theta_{BDSO})$.*

An open question is how many such critical points exist. A starting point is to consider that there are $C(N+H, H) \triangleq (N+H)!/N!H!$ possible partitions of the data. Not every such partition will admit a piecewise-OLS solution which is also continuous, and it is difficult to analytically characterize such solutions, so we resort to simulation and find a lower bound that suggests the number of critical points grows at least polynomially in $N$ and $H$ (Figure 7).

Using Theorem 3, we can characterize growth of global minima in the overparameterized case. Call a partition $\Pi$ *lonely* if each piece $\pi_p$ contains at most one datapoint. Then, we can prove the following results:

**Theorem 4.** *For any lonely partition $\Pi$, there are infinitely many parameter settings $\theta_{BDSO}$ that induce $\Pi$ and are global minima with $\tilde{\ell}(\theta_{BDSO}) = 0$.*

*Proof.* Note that each linear piece $p$ has two degrees of freedom (slope and intercept). By way of induction, start at (say) the left-most piece. If there is a datapoint in this piece, choose an arbitrary slope and intercept that goes through it; otherwise, choose an arbitrary slope and intercept. At each subsequent piece, we can use one degree of freedom to ensure continuity with the previous piece, and use one degree of freedom to match the data (if there is any). $\square$

**Remark 1.** *Suppose that the $H$ breakpoints are uniformly spaced and that the $N$ data points are uniformly distributed within the region of breakpoints. Then in the overparametrized regime $H \geq \alpha N^2$ for some constant $\alpha > 1$, the induced partition $\Pi$ is lonely with high probabililty $1 - e^{-N^2/(H+1)} = 1 - e^{-1/\alpha}$. Furthermore, the total number of lonely partitions, and thus a lower bound on the total number of global minima of $\tilde{\ell}$ is $\binom{H+1}{N} = O(N^{\alpha N})$*

Thus, with only order $N^2$ units, we can almost guarantee lonely partitions, where the piecewise OLS solution on these lonely paratitions will be the global optimum. Note how simple and transparent the function space explanation is for why overparametrization makes optimization easy, as compared to the weight space explanation (Arora et al., 2019), requiring order $N^7$ units.

## 2.4 Generalization: Implicit Regularization via Delta-Slope Parametrization

The above sections argue that overparameterization leads to a flatter initial function approximation, and an easier time reaching a global minima over the training data. However, neural networks also exhibit unreasonably high generalization performance, which must be due to *implicit* regularization, since the effect is independent of loss function. Here we provide an argument that overparameterization directly leads to this implicit regularization, due to the increasing flatness of the initialization and the non-locality of the delta-slope parameters.

Consider a dataset like that shown in Figure 8 with a data gap between regions of two continuous functions $f_L, f_R$ and consider a breakpoint $i$ with orientation $s_i$ in the gap. Starting with a flat initialization, the dynamics of the $i$-th delta-slope are $\dot{\mu}_i(t) = -\langle \hat{\epsilon}(t) \odot \mathbf{a}_i(t), \mathbf{x} \rangle + \beta_i(t) \langle \hat{\epsilon}(t) \odot \mathbf{a}_i(t), \mathbf{1} \rangle \triangleq r_{2,s_i}(t) + r_{3,s_i}(t)\beta_i(t)$ where $r_{2,s}(t), r_{3,s}(t)$ are the (negative) net correlation and residual on the active side of $i$, in this case including data from the function $f_{s_i}$ but not $f_{-s_i}$. Note that the both terms of the gradient $\dot{\mu}_i$ have a weak dependence on $i$ through the orientation $s_i$, and the second term additionally depends on $i$ through $\beta_i(t)$. Thus the vector of delta-slopes with orientation $s$ evolves according to $\dot{\boldsymbol{\mu}}_s = r_{2,s}(t)\mathbf{1} + r_{3,s}(t)\boldsymbol{\beta}_s$. Now consider the regime of overparametrization $H \gg N$. It will turn out to be identical to taking a continuum limit $H \to \infty$ yielding $\mu_i/(\beta_i - \beta_{i-1}) \to \mu(x,t) \triangleq \hat{f}''(x,t)$, the curvature of the approximation (the discrete index

$i$ has become a continuous index $x$) and $\dot{\beta}_i(t) \to 0$ (following from Theorem 5, multiplying $\dot{\beta}_i(t)$ by $v_i(t)/w_i(t)$ and factoring out $\mu_i(t) \to 0$). Integrating the dynamics $\dot{\mu}_s(x,t) = r_{2,s}(t) + r_{3,s}(t)x$ over all time yields $\mu_s(x, t = \infty) = \mu_s(x, t = 0) + R_{2,s}^* + R_{3,s}^* x$, where the curvature $\mu_s(x, t = 0) \approx 0$ (Section 3) and $R_{j,s}^* \triangleq \int_0^\infty dt' r_{j,s}(t') < \infty$ (convergence of residuals $\epsilon_n(t)$ and immobility of breakpoints $\dot{\beta}_i(t) = 0$ implies convergence of $r_{j,s}(t)$). Integrating over space twice(from $x' = \xi_s$ to $x' = x$) yields a cubic spline $\hat{f}(x,t) = c_{0,s} + c_{1,s}(x - \xi_s) + c_{2,s}(x - \xi_s)_s^2/2! + c_{3,s}(x - \xi_s)_s^3/3!$, where $c_{0,s}, c_{1,s}$ are integration constants determined by the per-piece boundary conditions (PBCs) $\hat{f}(x = \xi_s, t = \infty) \triangleq \sum_s \hat{f}_s(x = \xi_s) = f(x = \xi_s)$ and $\hat{f}'(x = \xi_s, t = \infty) \triangleq \sum_s \hat{f}_s'(x = \xi_s) = f'(x = \xi_s)$, thus matching the 0-th and 1st derivatives at the gap endpoints. The other two coefficients $c_{k,s} \triangleq R_{k,s}^*, k \in \{2, 3\}$ and serve to match the 2nd and 3rd derivatives at the gap endpoints. Clearly, matching the training data only requires the two parameters $c_{0,s}, c_{1,s}$; and yet, surprisingly, two unexpected parameters $c_{2,s}, c_{3,s}$ emerge that endow $\hat{f}$ with smoothness in the data gap, despite the loss function not possessing any explicit regularization term. Tracing back to find the origin of these smoothness-inducing terms, we see that they emerge as a consequence of (i) the smoothness of the initial function and (ii) the active half space structure, which in turn arises due to the discrete curvature-based (delta-slope) parameterization. Stepping back, the ReLU net parameterization is a discretization of this underlying continuous 2nd-order ordinary differential equation. In Section 3 we conduct experiments to test this theory.

## 3 EXPERIMENTS

**Breaking Bad: Breakpoint densities that are mismatched to function curvature makes optimization difficult** We first test our initialization theory against real networks. We initialize fully-connected ReLU networks of varying depths, according to the popular He initializations (He et al., 2015). Figure 1 shows experimentally measured densities of breakpoints and delta-slopes. Our theory matches the experiments well. The main points to note are that: (i) breakpoints are indeed more highly concentrated around the origin, and that (ii) as depth increases, delta-slopes have lower variance and thus lead to even flatter initial functions. We next ask whether the standard initializations will experience difficulty fitting functions that have significant curvature away from the origin (e.g. learning the energy function of a protein molecule). We train ReLU networks to fit a periodic function ($\sin(x)$), which has high curvature both at and far from the origin. We find that the standard initializations do quite poorly away from the origin, consistent with our theory that breakpoints are essential for modeling curvature. Probing further, we observe empirically that breakpoints cannot migrate very far from their initial location, even if there are plenty of breakpoints overall, leading to highly suboptimal fits. We additionally show (see Appendix A.6 for details) that breakpoint distributions change throughout training to more accurately match the ground truth curvature. In order to prove that it is indeed the breakpoint density that is causally responsible, we attempt to rescue the poor fitting by using a simple data-dependent initialization that samples breakpoints uniformly over the training data range $[x_{min}, x_{max}]$, achieved by exploiting Eq. (2).

We train shallow ReLU networks on training data sampled from a sine and a quadratic function, two extremes on the spectrum of curvature. The data shows that uniform breakpoint density rescues bad fits in cases with significant curvature far from the origin, with less effect on other cases, confirming the theory. We note that this could be

| Init | Sine | Quadratic |
|---|---|---|
| Standard | $4.096 \pm 2.25$ | $.1032 \pm 0404$ |
| Uniform | $2.280 \pm .457$ | $.1118 \pm .0248$ |

Table 1: Test loss for standard vs uniform breakpoint initialization, on sine and quadratic $\frac{x^2}{2}$

a potentially useful data-dependent initialization strategy, one that can scale to high dimensions, but we leave this for future work.

**Explaining and Quantifying the Suboptimality of Gradient Descent.** The suboptimality seen above begs a larger question: under what conditions will GD be successful? Empirically, it has been observed that neural nets must be massively overparameterized (relative to the number of parameters needed to express the underlying function), in order to ensure good training performance. Our theory provides a possible explanation for this phenomenon: if GD cannot move breakpoints too far from their starting point, then one natural strategy is to sample as many breakpoints as possible everywhere, allowing us to fit an arbitrary $f$. The downside of this strategy is that many breakpoints will add little value. In order to test this explanation and, more generally, understand the root causes

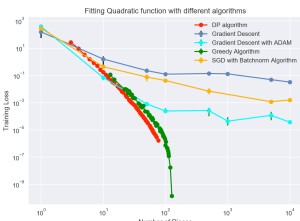 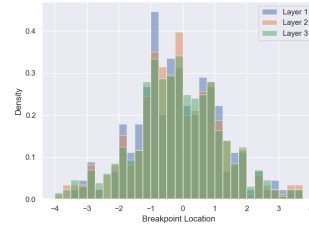 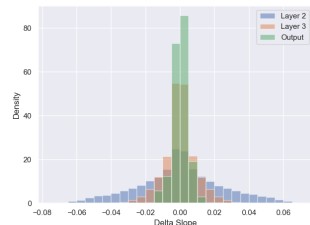

Figure 1: Left: Training loss vs number of pieces ($\propto$ number of parameters) for various algorithms fitting a CPWL function to a quadratic. Middle: Breakpoint distribution for a He initialization across a 3 layer network. Right: Delta-slope distribution for a He initialization across a 3 layer network.

| L | Sine | 5 piece poly | Sawtooth | Arctan | Exponential | Quadratic |
|---|------|--------------|----------|--------|-------------|-----------|
| 1 | $40 \pm 0$ | $40 \pm 0$ | $40 \pm 0$ | $40 \pm 0$ | $40 \pm 0$ | $40 \pm 0$ |
| 2 | $55.5 \pm 2.9$ | $52 \pm 1.414$ | $50 \pm .7$ | $49.25 \pm 3.3$ | $51.25 \pm 6.1$ | $49.25 \pm 4.5$ |
| 4 | $68 \pm 3.1$ | $57.25 \pm 6.8$ | $48.5 \pm 2.5$ | $42.5 \pm 4.8$ | $40.25 \pm 3.9$ | $40.25 \pm 3.3$ |
| 5 | $62.25 \pm 15.1$ | $49 \pm 3.5$ | $44.5 \pm 5.1$ | $38 \pm 5.1$ | $33.75 \pm 1.1$ | $31.5 \pm 1.7$ |

Table 2: Comparison of the number of pieces induced in a network of up to depth 5, with 40 units evenly distributed across layers, trained to fit varying target functions.

of the GD's difficulty, we focus on the case of a fully connected shallow ReLU network. A univariate input (i) enables us to use our theory, (ii) allows for visualization of the entire learning trajectory, and (iii) enables direct comparison with existing globally (near-)optimal algorithms for fitting PWL functions. The latter include the Dynamic Programming algorithm (DP, (Bai & Perron, 1998)), and a very fast greedy approximation known as Greedy Merge (GM, (Acharya et al., 2016)). How do these algorithms compare to GD, across different target function classes, in terms of training loss, and the number of pieces/hidden units? We use this metric for the neural network as well, rather than the total number of trainable parameters.

Taking the functional approximation view allows us to directly compare neural network performance to these PWL approximation algorithms. For a quadratic function (e.g. with high curvature, requiring many pieces), we find that the globally optimal DP algorithm can quickly reduce training error to near 0 with order 100 pieces. The GM algorithm, a relaxation of the DP algorithm, requires slightly higher pieces, but requires significantly less computational power. On the other hand all variants of GD (vanilla, Adam, SGD w/ BatchNorm) all require far more pieces to reduce error below a target threshold, and may not even monotonically decrease error with number of pieces. Interestingly, we observe a strict ordering of optimization quality with Adam outperforming BatchNorm SGD outperforming Vanilla GD. These results (Figure 1) show how inefficient GD is with respect to (functional) parameters, requiring orders of magnitude more for similar performance to exact or approximate PWL fitting algorithms.

**Learned Expressivity is not Exponential in Depth.** In the previous experiment, we counted the number of linear pieces in the CPWL approximation as the number of parameters, rather than the number of weights. Empirically, we know that the greatest successes have come from *deep* learning. This raises the question: how does the depth of a network affect its expressivity (as measured in the number of pieces)? Theoretically, it is well known that maximum expressivity increases exponentially with depth, which, in a deep ReLU neural network, means an exponential increase in the number of linear pieces in the CPWL approximation. Thus, theoretically the main power of depth is that it allows for more powerful function approximation relative to a fixed budget of parameters compared to a shallow network. However, recent work (Hanin & Rolnick, 2019) has called this into question, finding that in realistic networks expressivity does not scale exponentially with depth. We perform a similar experiment here, asking how the number of pieces in the CPWL function approximation of a deep ReLU network varies with depth.

The results in Table 2 clearly show that the number of pieces does not exponentially scale with depth. In fact, we find that depth only has a weak effect overall, although more study is needed to determine exactly what effect depth has on the number and variability of pieces. These results lend more support to the recent findings of (Hanin & Rolnick, 2019), and of taking a functional view

| Function | Shallow | Spiky Shallow | Deep | Spiky Deep |
|---|---|---|---|---|
| Sine | $42.95 \pm 6.406$ | $157.5 \pm 60.27$ | $31.48 \pm 7.078$ | $122.0 \pm 128.2$ |
| Arctan | $.01252 \pm .07650$ | $2.499 \pm 1.257$ | $0.9795 \pm 0.9355$ | $32.57 \pm 26.10$ |
| Sawtooth | $156.9 \pm 12.45$ | $150.1 \pm 61.48$ | $148.1 \pm 8.755$ | $198.0 \pm 170.9$ |
| Cubic | $3.608 \pm 1.683$ | $136.7 \pm 124.1$ | $56.77 \pm 98.91$ | $191.6 \pm 114.1$ |
| Quadratic | $3.559 \pm 4.553$ | $150.6 \pm 49.00$ | $1.741 \pm 1.296$ | $46.02 \pm 19.42$ |
| Exp | $.6509 \pm .5928$ | $181.1 \pm 75.36$ | $1.339 \pm 1.292$ | $54.50 \pm 37.77$ |

Table 3: Comparison of testing loss (generalization ability) of various network shallow and deep networks with a standard vs 'spiky' initialization

of measuring parameterization. Intriguingly, variability in the number of pieces appears to increase with depth. From the functional approximation, we know that a unit induces breakpoints only when the ReLU function applied to the unit's input has zero crossings. In layer one, this happens exactly once per unit as the input to each ReLU is just a line over the input space. In deeper layers, the function approximation is learned, allowing for a varying number of new breakpoints. Given our previous results on the flatness of the standard initializations, this will generally only happen once per unit, implying that the number of pieces will strongly correlate with the number of units at initialization.

**Depth helps with Optimization by enabling the Creation, Annihilation and Mobility of Breakpoints.** If depth does not strongly increase expressivity, then it is natural to ask whether its value lies with optimization. In order to test this, we examine how the CPWL function approximation develops in each layer during learning, and how it depends on the target function. A good fit requires that breakpoints accumulate at areas of higher curvature in the training data, as these regions require more pieces. We argue that the deeper layers of a network help with this optimization procedure, allowing the breakpoints more mobility as well as the power to create and annihilate breakpoints.

One key difference between the deeper layers of a network and the first layer is the ability for a single unit to induce multiple breakpoints. As these units' inputs change during learning, the number of breakpoints induced by deeper units in a network can vary, allowing for another degree of freedom for the network to optimize. Through the functional parameterization of the hidden layers, these "births and deaths" of breakpoints can be tracked as changes in the number of breakpoints induced per layer. Another possible explanation for the value added of depth is breakpoint mobility, or that breakpoints in deeper layers can move more than those in shallow layers. We run experiments comparing how the velocity and number of induced breakpoints varies between layers of a deeper network.

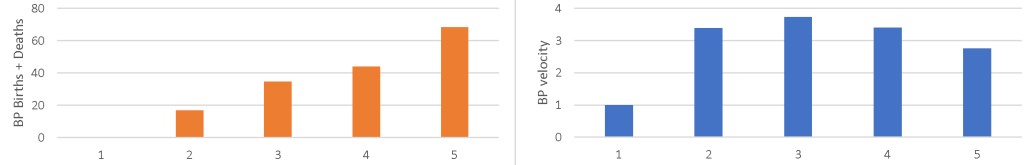

Figure 2: Total changes in number of breakpoints induced and average velocity of breakpoints relative to the first layer in each layer of a five layer ReLU network

Figure 2 shows the results. The number of breakpoints in deeper layers changes more often than in shallow layers. The breakpoint velocity in deeper layers is also higher than the first layer, although not monotonically increasing. Both of these results provide support for the idea that later layers help significantly with optimization and breakpoint placement, even if they do not help as strongly with expressivity.

Note that breakpoints induced by a layer of the network are present in the basis functions of all deeper layers. Their functional approximations thus become more complex with depth. However the roughness of the basis functions at initialization in the deeper layers is lower than that of the shallow layers. But, as the network learns, for complex functions most of the roughness is in the later layers as seen in Figure 3 (right).

**Generalization: Implicit Regularization emerges from Flat Init and Curvature-based Parametrization.** The experiments above show that the functional view can give us a new per-

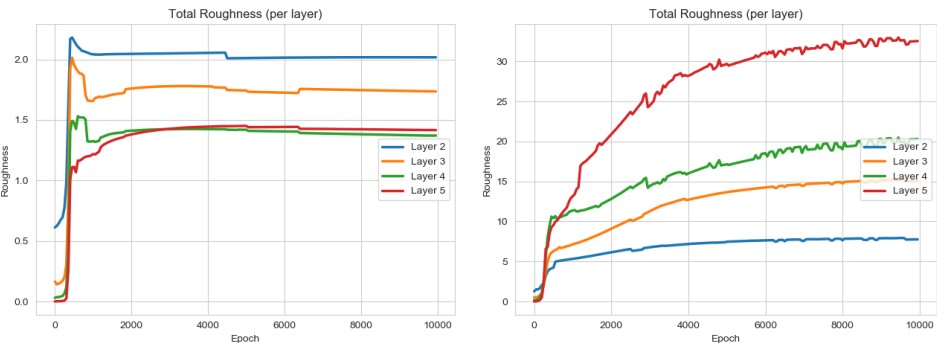

Figure 3: Roughness (summed by layer) during training for a 5 layer ReLU network with 8 units per hidden layer, learning the quadratic function $x^2/2$ (left) and the periodic function $\sin(x)$ (right)

spective on how depth and parameterization affect the training of neural networks. One of the most useful and perplexing properties of deep neural networks has been that, in contrast to other high capacity function approximators, overparameterizing a neural network does not tend to lead to excessive overfitting (Savarese et al., 2019). Where does this generalization power come from? Much recent work (Neyshabur et al., 2018; 2015) has argued that it comes from an implicit regularization inherent in the optimization algorithm itself (i.e. SGD). In contrast, for the case of shallow and deep univariate fully connected ReLU nets, we provide causal evidence that it is due to the specific, very flat CPWL initialization induced by common initialization methods. In order to test this in both shallow and deep ReLU networks, we compare training with the standard flat initialization to a 'spiky' initialization.

For a shallow ReLU network, we can test a 'spiky' initialization by exactly solving for network parameters to generate a given arbitrary CPWL function. This network initialization is then compared against a standard initialization, and trained against a smooth function with a small number of training data points. Note that in a 1D input space we need a small number of training data points to create a situation similar to that of the sparsity caused by high dimensional input, and to allow for testing generalization between data points. We find that both networks fit the training data near perfectly, reaching a global minima of the training loss, but that the 'spiky' initialization has much worse generalization error (Table 3). Visually, we find that the initial 'spiky' features of the starting point CPWL representation are preserved in the final approximation of the smooth target function (Figures 4 and 6). For a deep ReLU network, it is more difficult to exactly solve for a 'spiky' initialization. Instead, we train a network to approximate an arbitrary CPWL function, and call those trained network parameters the 'spiky' initialization. Once again, the 'spiky' initialization has near identical training performance, hitting all data points, but has noticeably worse generalization performance.

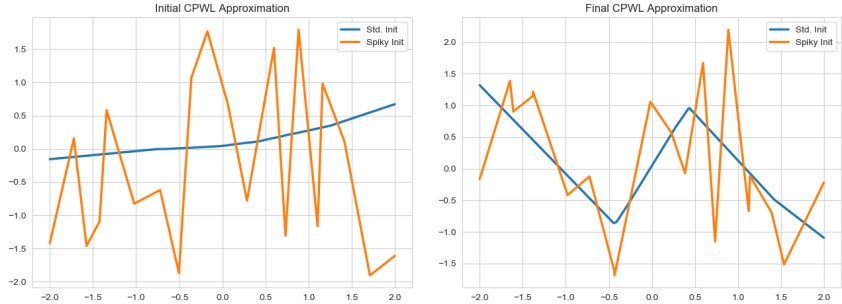

Figure 4: 'Spiky' (orange) and standard initialization (blue), compared before (left) and after (right) training. Note both cases had similar, very low training set error.

It appears that generalization performance it not automatically guaranteed by GD, but instead due to the flat initializations which are then *preserved* by GD. 'Spiky' initializations also have their (higher) curvature preserved by GD. This idea makes sense, as generalization depends on our target function smoothly varying, and a smooth approximation is promoted by a smooth initialization.

**Smoothness in Data Gaps increases with Hidden Units and Decreases with Initial Weight Variance.** Our last experiment examines how smoothness (roughness) depends on the number of units, particularly in the case where there are large gaps in the training data. We use a continuous and discontinuous target function (shown in Figure 8). We trained shallow ReLU networks with varying width $H$ and initial weight variance $\sigma_w$ on these training data until convergence, and measured the total roughness of resulting CPWL approximation in the data gaps.

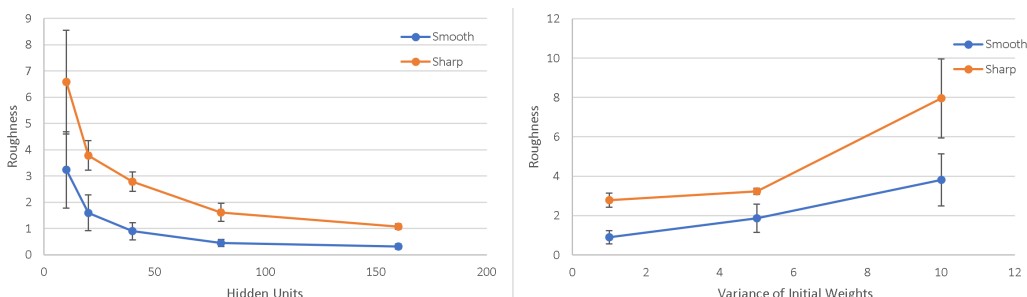

Figure 5: Roughness vs. Width (left) and the variance of the initialization (right) for both data gap cases shown in Figure 8. Each data point is the result of averaging over 4 trials trained to convergence.

Figure 5 shows that roughness in the data gaps decreases with width and increases with initial weight variance, confirming our theory. A spiky (and thus rougher) initialization leads to increased roughness at convergence as well, lending support to the idea that roughness in data gaps can be 'remembered' from initialization. On the other hand, higher number of pieces spreads out the curvature work over more units, leading to smaller overall roughness. Taken together, our experiments indicate that smooth, flat initialization is partly (if not wholly) responsible for the phenomenon of implicit regularization in univariate fully connected ReLU nets, and that increasing overparameterization leads to even better generalization.

**Conclusions.** We show in this paper that examining deep networks through the lens of function space can enabled new theoretical and practical insights. We have several interesting findings: the value of depth in deep nets seems to be less about expressivity and more about learnability, enabling GD to finding better quality solutions. The functional view also highlights the importance initialization: a smooth initial approximation seems to encourage a smoother final solution, improving generalization. Fortunately, existing initializations used in practice start with smooth initial approximations, with smoothness increasing with depth. Analyzing the loss surface for a ReLU net in function space gives us a surprisingly simple and transparent view of the phenomenon of overparameterization: it makes clear that increasing width relative to training data size leads w.h.p. to lonely partitions of the data which are global minima. Function space shows us that the mysterious phenomenon of implicit regularization may arise due to a hidden 2nd order differential equation that underlies the ReLU parameterization. In addition, this functional lens suggests new tools, architectures and algorithms. Can we develop tools to help understand how these CPWL functions change across layers or during training? Finally, our analysis shows that bad local minima are often due to breakpoints getting trapped in bad local minima: Can we design new learning algorithms that make *global* moves *in the BDSO parameterization* in order to avoid these local minima?

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

# A    EXPERIMENTAL DETAILS

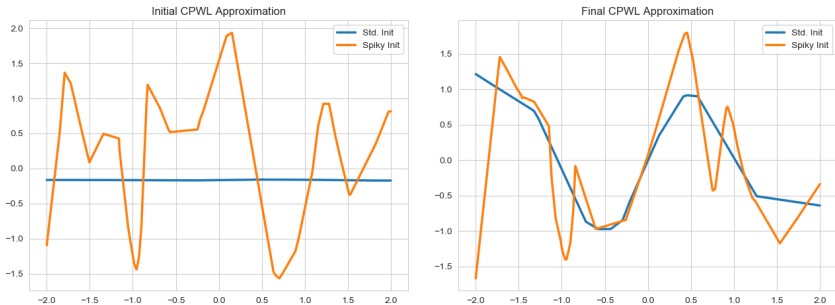

Figure 6: 'Spiky' (orange) and standard initialization (blue), compared before training (left) and post-training (right) using a deep network

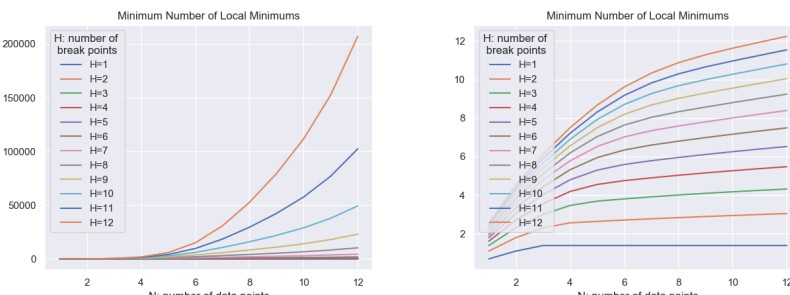

Figure 7: Growth in the (minimum) amount of local minima, as a function of the number of breakpoints and data points. Right plot is identical, but with log scaling

## A.1    UNIFORM INITIALIZATION

Trained on a shallow, 21 unit FC ReLU network. Trained on function over the interval [-2,2]. Learning rate = 5e-5, trained via GD over 10000 epochs. Compared against pytorch default of He initialization. Training data sampled uniformly every .01 of the target interval. Each experiment was run 5 times, with results reported as mean $\pm$ standard deviation. Breakpoints y values were taken from the original standard initialization for the uniform initialization plus a small random noise term $N(0,.01)$, making initial condition within the target interval nearly identical.

## A.2    ROUGHNESS BY LAYER PLOTS

Trained on a deep, 5 layer network, with 4 hidden layers of width 8. Trained on function over the interval [-2,2]. Learning rate = 1e-4, trained via GD over 10000 epochs, with roughness measured every 50 epochs. Roughness per layer was summed over all units within that layer.

## A.3    SPIKY INITIALIZATION PLOTS

Shallow version trained on a 21 unit FC ReLU Network. Deep version trained on a deep, 5-layer network with 4 hidden layers of width 8. In both cases, the 'spiky' initialization was a 20 - breakpoint CPWL function, with $y_n \sim \text{Uniform}([-2, 2])$. In the deep case, the spiky model was initialized with the same weights as the non-spiky model, and then pre-trained for 10,000 epochs to fit the CPWL. After that, gradient descent training proceeded on both models for 20,000 epochs, with all training having learning rate 1e-4. Training data was 20 random points in the range [-2,2], while the testing

data (used to measure generalization) was spaced uniformly at every $\Delta x = .01$ of the target interval of the target function.

In the shallow case, there was no pre-training, as the 'spiky' model was directly set to be equal to the CPWL. In the shallow model, training occurred for 20,000 epochs. All experiment were run over 5 trials, and values in table are reported as mean $\pm$ standard deviation. Base shallow learning rate was 1e-4 using gradient descent method, with learning rate divided by 5 for the spiky case due to the initialization method generating larger weights. Despite differing learning rates, both models had similar training loss curves and similar final training loss values, e.g. for sine, final training loss was .94 for spiky and 1.02 for standard. Functions used were $\sin(x), \arctan(x)$, a sawtooth function from [-2,2] with minimum value of -1 at the endpoints, and 4 peaks of maximum value 1, cubic $\frac{x^3}{4} + \frac{x^2}{2} - \frac{x}{2}$, quadratic $\frac{x^2}{2}$, and $\exp(.5x)$ Note GD was chosen due to the strong theoretical focus of this paper - similar results were obtained using ADAM optimizer, in which case no differing learning rates were necessary.

### A.4 Breakpoints Induced by Deep Networks

We used networks with a total of $H = 40$ hidden units, spread over $L \in \{1, 2, 3, 4, 5\}$ hidden layers. Training data consiste of uniform samples of function over the interval $x \in [-3, 3]$. Learning rate $= 5 \cdot 10^{-5}$, trained via GD over $25,000$ epochs. The target functions tested were $\sin(\pi x)$, a 5-piece polynomial with maximum value of 2 in the domain $[-3, 3]$, a sawtooth with period 3 and amplitude 1, $\arctan(x)$, $\exp(x)$, and $\frac{1}{9}x^2$. Each value in the table was the average of 5 trials.

### A.5 Breakpoint Mobility in Deep Networks

We use a deep, 6-layer network, with 5 hidden layers of width 8. Training data consists of the 'smooth' and 'sharp' functions over the interval $x \in [-3, 3]$. Learning rate = 5e-5, trained via GD until convergence, where convergence was defined as when the loss between two epochs changed by less than $10^{-8}$. Breakpoints were calculated every 50 epochs. The velocity of breakpoints was then calculated, and the values seen in the figure are normalized to the velocity of the first layer.

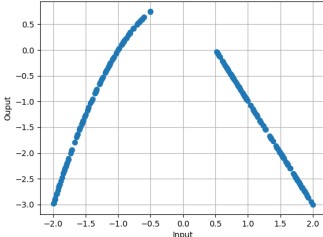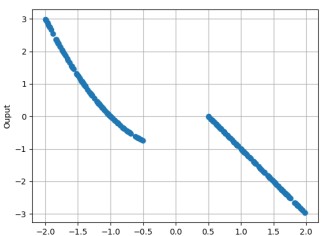

Figure 8: Training data sampled from two ground truth functions, one smoothly (left) and the other sharply (right) discontinuous, each with a data gap at $[-0.5, 0.5]$.

### A.6 Breakpoint Distributions

Various function classes were trained until convergence on a depth 1 or 4 ReLU network, with 500 total units distributed evenly across layers. Initial and final breakpoint distributions were measured using a kernel density estimate, and compared with the underlying curvature (absolute value of 2nd derivative) of the ground truth function. The cubic spline was a cubic spline fit to a small number of arbitrary data points.

Table 4 shows that the breakpoint densities moved over training to become more correlated with the underlying curvature of the ground truth function. This effect was more pronounced with depth. In certain very simple functions (e.g. $x^2$ or $exp(x)$, not shown), a failure case emerged where there was no real change in correlation over training. Diagnostics appeared to show this was due to the function being so simple as to train almost instantaneously in our overparameterized network, meaning breakpoints had no time to move. Figure 9 shows what happens to the breakpoint densities

| Depth/Type | Sine | $x^3$ | $Sin(\frac{x^2}{2})$ | Cubic Spline |
|---|---|---|---|---|
| 1 - Initial | -0.0770 | -0.904 | -0.754 | -0.164 |
| 1 - Final | 0.324 | -0.891 | -0.592 | 0.289 |
| 4 - Initial | -0.171 | -0.900 | -0.752 | -0.133 |
| 4 - Final | 0.494 | 0.688 | -0.212 | 0.798 |
| 1 - $\Delta$ | 0.401 | 0.0130 | 0.162 | 0.452 |
| 4 - $\Delta$ | 0.665 | 1.59 | 0.540 | 0.931 |

Table 4: Top: Correlation of the BP distribution before and after training for depth 1 and 4 networks across function classes. Bottom : Change in correlation over training

over training - in the shallow case, they are more constrained by the initial condition, and continue to have a higher density near the origin even when not necessary or appropriate.

## B    MORE DETAILS ON BREAKPOINTS FOR DEEP RELU NETS

Each neuron of the second hidden layer receives as input the result of a CPWL function $z_i^{(2)}(x)$ as defined above. The output of this function is then fed through a ReLU, which has two implications: first, every zero crossing of $z_i^{(2)}$ is a breakpoint of $x_i^{(2)}$; second, any breakpoints $\beta_j^{(1)}$ of $z_i^{(2)}$ such that $z_i^{(2)}(\beta_j^{(1)}) < 0$ will not be breakpoints of $x_i^{(2)}$. Importantly, the number of breakpoints in $g_\theta(x)$ is now a function of the parameters $\theta$, rather than equal to fixed $H$ as in the $L = 1$ case; in other words, breakpoints can be dynamically created and annihilated throughout training. This fact will have dramatic implications when we explore how gradient descent optimizes breakpoints in order to model curvature in the training data (see Section 3). But first, due to complexities of depth, we must carefully formalize the notion of a breakpoint for a deep network.

**Definition 1.** $\beta_i^{(\ell)}$ *is a* breakpoint induced by neuron $i$ in layer $\ell$ if $z_i^{(\ell)}(\beta_i^{(\ell)}) = 0$. *Since the function* $z_i^{(\ell)}(\cdot)$ *is nonlinear, neuron* $i$ *may induce multiple breakpoints, which we denote* $\beta_{i,k}^{(\ell)}$. *A breakpoint* $\beta_{i,k}^{(\ell)}$ *is* active *if there exists some path* $\pi$ *through neuron* $i$ *such that for all other neurons* $j \neq i \in \pi$, $z_j^{(\ell(j))} > 0$, *i.e.* $\hat{a}_j(x) = 1$. *If two neurons* $i$ *and* $j$ *in layers* $\ell$ *and* $\ell'$ *induce the same breakpoint(s),* $\beta_{i,k}^{(\ell)} = \beta_{j,k'}^{(\ell')}$, *then both are referred to as* degenerate *breakpoints.*

Let $\hat{a}_\pi(x) = \prod_{i \in \pi} \hat{a}_i$. Then, $\beta_i^{(\ell)}$ is active iff there exists some path $\pi$ such that $\hat{a}_\pi$ is discontinuous at $x = \beta_i^{(\ell)}$. Thus, $g_\theta(x)$ is non-differentiable at $x$ if $x = \beta_i^{(\ell)}$ for some $(\ell, i)$. If no degenerate breakpoints exist, then the converse also holds. (If there do exist degenerate breakpoints $\beta_i^{(\ell)}$ and $\beta_j^{(\ell')}$, then it is possible that $\mu_i^{(\ell)} = -\mu_j^{(\ell')}$, i.e. the changes in slope cancel out and $g_\theta(x)$ remains linear and differentiable.)

## C    PROOFS OF THEORETICAL RESULTS

### C.1    REPARAMETRIZATION FROM RELU NETWORK TO PIECEWISE LINEAR FUNCTION

*Proof of Equations* (2) *and* (3)*:*

$$
\begin{aligned}
\hat{f}_{\theta,H}(x) &= \sum_{i=1}^{H} v_i \phi(w_i x + b_i) \\
&= \sum_{i=1}^{H} v_i (w_i x + b_i) [\![ w_i x + b_i > 0 ]\!] \\
&= \sum_{i=1}^{H} v_i w_i (x - \beta_i) \begin{cases} [\![ x > \beta_i ]\!] & w_i > 0 \\ [\![ x < \beta_i ]\!] & w_i < 0 \end{cases} \quad \text{where } \beta_i \triangleq -\frac{b_i}{w_i} \\
&= \sum_{i=1}^{H} \mu_i (x - \beta_i) \begin{cases} [\![ x > \beta_i ]\!] & w_i > 0 \\ [\![ x < \beta_i ]\!] & w_i < 0 \end{cases} \quad \text{where } \mu_i \triangleq v_i w_i
\end{aligned}
$$

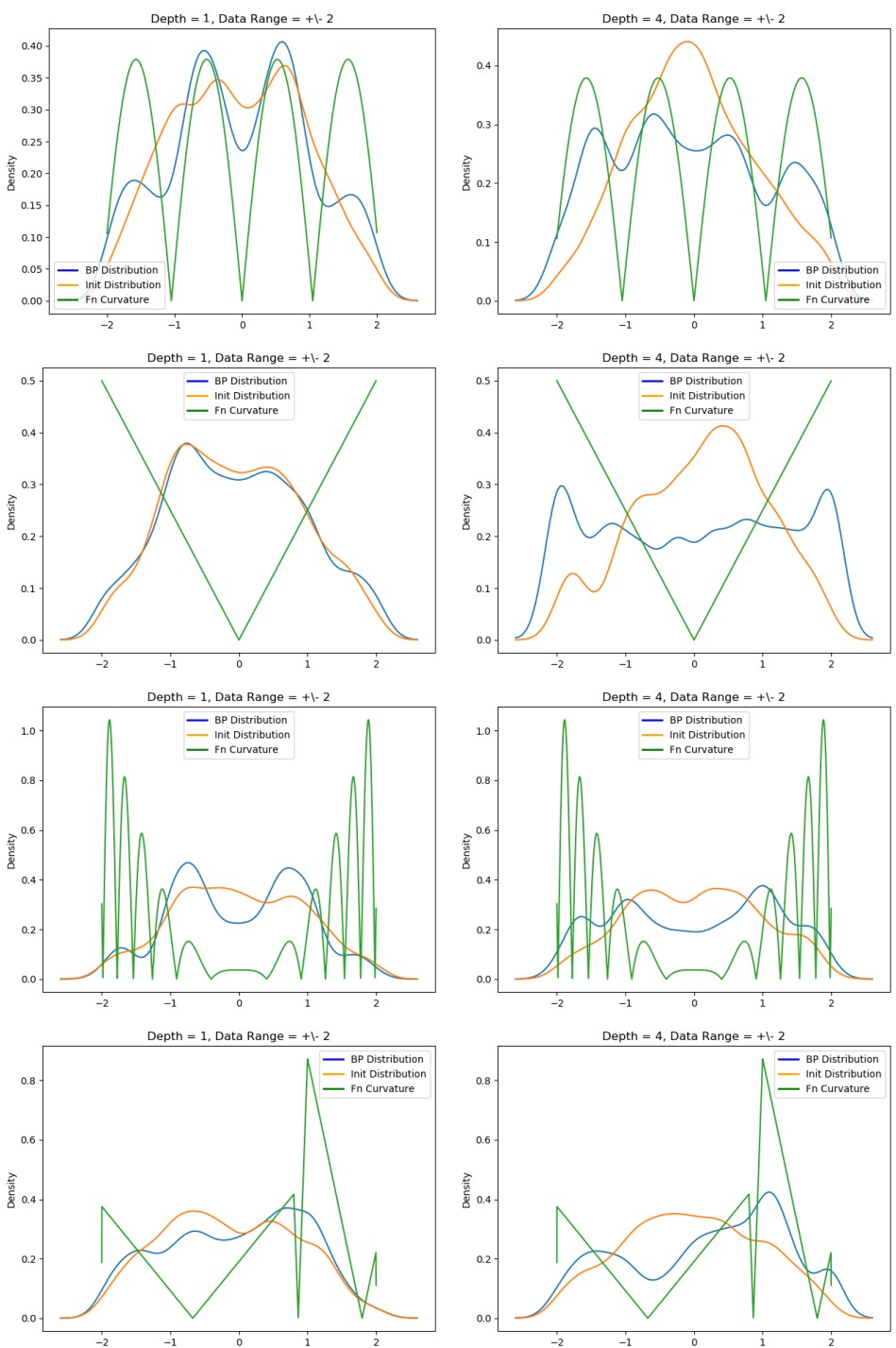

Figure 9: Shallow (left) and deep (right) plots of initial and final breakpoint distributions, along with the underlying true curvature of the functions (top to bottom) $sin(x)$, $x^3$, $sin(\frac{x^2}{2})$, and a cubic spline of a few arbitrary data points

This gives us Equation (2), as desired. Let the subscripts $p$, $q$ denote the parameters sorted by $\beta_p$ value. In this setting, let $\beta_0 \triangleq -\infty$, and $\beta_{H+1} \triangleq \infty$. Then,

$$
\begin{aligned}
&= \sum_{p=0}^{H} \left( \sum_{\substack{q=1 \\ w_q>0}}^{p} \mu_q(x-\beta_q) + \sum_{\substack{q=p+1 \\ w_q<0}}^{H} \mu_q(x-\beta_q) \right) [\![\beta_p \le x < \beta_p]\!] \\
&= \sum_{p=0}^{H} \left( \sum_{\substack{q=1 \\ w_q>0}}^{p} (\mu_q x - \mu_q\beta_q) + \sum_{\substack{q=p+1 \\ w_q<0}}^{H} (\mu_q x - \mu_q\beta_q) \right) [\![\beta_p \le x < \beta_{p+1}]\!] \\
&= \sum_{p=0}^{H} \left( \sum_{\substack{q=1 \\ w_q>0}}^{p} \mu_q x - \sum_{\substack{q=1 \\ w_q>0}}^{p} \mu_q\beta_q + \sum_{\substack{q=p+1 \\ w_q<0}}^{H} \mu_q x - \sum_{\substack{q=p+1 \\ w_q<0}}^{H} \mu_q\beta_q \right) [\![\beta_p \le x < \beta_{p+1}]\!] \\
&= \sum_{p=0}^{H} \left( \left( \sum_{\substack{q=1 \\ w_q>0}}^{p} \mu_q \right) x - \sum_{\substack{q=1 \\ w_q>0}}^{p} \mu_q\beta_q + \left( \sum_{\substack{q=p+1 \\ w_q<0}}^{H} \mu_q \right) x - \sum_{\substack{q=p+1 \\ w_q<0}}^{H} \mu_q\beta_q \right) [\![\beta_p \le x < \beta_{p+1}]\!] \\
&= \sum_{p=0}^{H} \left( \underbrace{\left( \sum_{\substack{q=1 \\ w_q>0}}^{p} \mu_q + \sum_{\substack{q=p+1 \\ w_q<0}}^{H} \mu_q \right)}_{\triangleq m_p} x - \underbrace{\left( \sum_{\substack{q=1 \\ w_q>0}}^{p} \mu_q\beta_q + \sum_{\substack{q=p+1 \\ w_q<0}}^{H} \mu_q\beta_q \right)}_{\triangleq \gamma_p} \right) [\![\beta_p \le x < \beta_{p+1}]\!] \\
&= \sum_{p=0}^{H} (m_p x - \gamma_p) [\![\beta_p \le x < \beta_{p+1}]\!]
\end{aligned}
$$

This gives us Equation (3), as desired. $\qquad\square$

## C.2 RANDOM INITIALIZATION IN FUNCTION SPACE

**Lemma 1.** *Suppose $(b_i, w_i, v_i)$ are initialized independently with densities $f_B(b_i)$, $f_W(w_i)$, and $f_V(v_i)$. Then, the density of $(\beta_i, \mu_i)$ is given by*

$$
f_{\beta,\mu}(\beta_i, \mu_i) = \int_{-\infty}^{\infty} f_B(\beta_i u) f_W(u) f_V(\frac{\mu_i}{u}) \, \mathrm{d}u \,.
$$

*Proof.* Suppose $(b_i, w_i, v_i)$ are initialized i.i.d. from a distribution with density $f_{B,W,V}(b_i, w_i, v_i)$. Then, we can derive the density of $(\beta_i, \mu_i)$ by considering the invertable continuous transformation given by $(\beta_i, \mu_i, u) = g(b_i, w_i, v_i) = (b_i/w_i, v_i|w_i|, w_i)$, where $g^{-1}(\beta_i, \mu_i, u) = (\beta_i u, u, \mu_i/|u|)$. The density of $(\beta_i, \mu_i, u)$ is given by $f_{B,M,V}(\beta_i u, u, \mu_i/|u|)|J|$, where $J$ is the Jacobian determinant of $g^{-1}$. Then, we have $J = -\operatorname{sgn} w_i$ and $|J| = 1$. The density of $(\beta_i, \mu_i)$ is then derived by integrating out the dummy variable $u$: $f_{\beta,\mu}(\beta_i, \mu_i) = \int_{-\infty}^{\infty} f_{B,W,V}(\beta_i u, u, \frac{\mu_i}{u}) \, \mathrm{d}u$. If $(b_i, w_i, v_i)$ are independent, this expands to $\int_{-\infty}^{\infty} f_B(\beta_i u) f_W(u) f_V(\frac{\mu_i}{u}) \, \mathrm{d}u$. $\qquad\square$

## C.3 GAUSSIAN INITIALIZATION IN FUNCTION

**Theorem 1(a).** *Consider a fully connected ReLU neural net with scalar input and output, and a single hidden layer of width $H$. Let the weights and biases be initialized randomly according to a zero-mean Gaussian or Uniform distribution. Then, under an independent Gaussian initialization,*

$$
p_{\beta,\mu}(\beta_i, \mu_i) = \frac{1}{2\pi\sigma_v \sqrt{\sigma_b^2 + \sigma_w^2\beta_i^2}} \exp\left[ -\frac{|\mu_i|\sqrt{\sigma_b^2 + \sigma_w^2\beta_i^2}}{\sigma_b\sigma_v\sigma_w} \right]
$$

*Proof.* Starting with Lemma 1,

$$
f_{\beta,\mu}(\beta, \mu) = \int_{-\infty}^{\infty} f_B(\beta_i u) f_W(u) f_V(\frac{\mu_i}{u}) \, \mathrm{d}u
$$

$$= \int_{-\infty}^{\infty} \frac{1}{\sqrt{2\pi\sigma_b^2}} e^{-\frac{(\beta u)^2}{2\sigma_b^2}} \frac{1}{\sqrt{2\pi\sigma_w^2}} e^{-\frac{u^2}{2\sigma_w^2}} \frac{1}{\sqrt{2\pi\sigma_v^2}} e^{-\frac{(\mu/u)^2}{2\sigma_v^2}} du$$

$$(\text{Sympy}) = \begin{cases} \dfrac{\exp\left[-\frac{\mu\sqrt{\sigma_b^2+\sigma_w^2(\beta)^2}}{\sigma_b\sigma_v\sigma_w}\right]}{2\pi\sigma_v\sqrt{\sigma_b^2+\sigma_w^2(\beta)^2}} & \mu > 0 \\ \text{unknown} & \text{otherwise} \end{cases}$$

but the integrand is even in $\mu$, giving

$$= \frac{\exp\left[-\frac{|\mu|\sqrt{\sigma_b^2+\sigma_w^2(\beta)^2}}{\sigma_b\sigma_v\sigma_w}\right]}{2\pi\sigma_v\sqrt{\sigma_b^2+\sigma_w^2(\beta)^2}}$$

$\square$

**Corollary 1(a).** *Consider the same setting as Theorem 1. In the case of an independent Gaussian initialization,*

$$p_\beta(\beta_i) = Cauchy\left(\beta_i; 0, \frac{\sigma_b}{\sigma_w}\right) = \frac{\sigma_b\sigma_w}{\pi\left(\sigma_w^2\beta_i^2+\sigma_b^2\right)}$$

$$p_\mu(\mu_i) = \frac{1}{2\pi\sigma_v\sigma_w} G_{0,2}^{2,0}\left(\frac{\mu_i^2}{4\sigma_v\sigma_w}\bigg| 0,0\right) = \frac{1}{\pi\sigma_v\sigma_w} K_0\left(\frac{|\mu_i|}{\sigma_v\sigma_w}\right)$$

$$p_{\mu|\beta}(\mu_i|\beta_i) = Laplace\left(\mu_i; 0, \frac{\sigma_b\sigma_v\sigma_w}{\sqrt{\sigma_b^2+\sigma_w^2\beta_i^2}}\right) = \frac{\sqrt{\sigma_b^2+\sigma_w^2\beta_i^2}}{2\sigma_b\sigma_v\sigma_w}\exp\left[-\frac{|\mu_i|\sqrt{\sigma_b^2+\sigma_w^2\beta_i^2}}{\sigma_b\sigma_v\sigma_w}\right],$$

*where $G_{pq}^{nm}(\cdot|\cdot)$ is the Meijer G-function and $K_\nu(\cdot)$ is the modified Bessel function of the second kind.*

*Proof.* Marginalizing out $\mu$ from the joint density in Sympy returns the desired $f_\beta(\beta)$ from above. Sympy cannot compute the other marginal, so we verify it by hand:

$$f_\mu(\mu) = \int_{-\infty}^{\infty} \frac{\exp\left[-\frac{|\mu|\sqrt{\sigma_b^2+\sigma_w^2\beta^2}}{\sigma_b\sigma_v\sigma_w}\right]}{2\pi\sigma_v\sqrt{\sigma_b^2+\sigma_w^2\beta^2}} dx = \frac{1}{2\pi\sigma_v}\int_{-\infty}^{\infty}\frac{\exp\left[-\frac{|\mu|\sqrt{\sigma_b^2+\sigma_w^2\beta^2}}{\sigma_b\sigma_v\sigma_w}\right]}{\sqrt{\sigma_b^2+\sigma_w^2\beta^2}}dx$$

$$\left(\phi(\beta) = \frac{\beta}{\sigma_w}\right) \qquad = \frac{1}{2\pi\sigma_v \underbrace{\sigma_w}_{\phi'(\beta)}}\int_{-\infty}^{\infty}\frac{\exp\left[\frac{-|\mu|\sqrt{\sigma_b^2+\beta^2}}{\sigma_b\sigma_v\sigma_w}\right]}{\sqrt{\sigma_b^2+\beta^2}}dx$$

from Gradshteyn & Ryzhik (2015), Eq. 3.462.20, we have

$$K_0(ab) = \int_0^{\infty}\frac{\exp\left(-a\sqrt{\beta^2+b^2}\right)}{\sqrt{\beta^2+b^2}}\,\mathrm{d}\beta \qquad [\operatorname{Re}a>0, \operatorname{Re}b>0]$$

$$(\text{integrand is even in }\beta) = \frac{1}{2}\int_{-\infty}^{\infty}\frac{\exp\left(-a\sqrt{\beta^2+b^2}\right)}{\sqrt{\beta^2+b^2}}\,\mathrm{d}\beta \qquad [\operatorname{Re}a>0, \operatorname{Re}b>0]$$

applying this with $a = \frac{|\mu|}{\sigma_b\sigma_v\sigma_w}$ and $b = \sigma_b$,

$$\frac{1}{2\pi\sigma_v\sigma_w}\int_{-\infty}^{\infty}\frac{\exp\left[\frac{-|\mu|\sqrt{\sigma_b^2+\beta^2}}{\sigma_b\sigma_v\sigma_w}\right]}{\sqrt{\sigma_b^2+\beta^2}}\,\mathrm{d}\beta = \frac{1}{\pi\sigma_v\sigma_w}K_0\left(\frac{|\mu|}{\sigma_v\sigma_w}\right)$$

as desired. We can then use these densities to derive the conditional:

$$f_\mu(\mu|\beta) = \frac{\sqrt{\sigma_b^2+\sigma_w^2(\beta)^2}\exp\left[-\frac{|\mu|\sqrt{\sigma_b^2+\sigma_w^2(\beta)^2}}{\sigma_b\sigma_v\sigma_w}\right]}{2\sigma_b\sigma_v\sigma_w}$$

$$= \text{Laplace}\left(\mu; 0, \frac{\sigma_b \sigma_v \sigma_w}{\sqrt{\sigma_b^2 + \sigma_w^2 (\beta)^2}}\right).$$

$\square$

## C.4 Uniform Initialization in Function Space

**Theorem 1(b).** *Consider a fully connected ReLU neural net with scalar input and output, and a single hidden layer of width $H$. Let the weights and biases be initialized randomly according to a zero-mean Gaussian or Uniform distribution. Then, under an independent Uniform initialization,*

$$p_{\beta,\mu}(\beta_i, \mu_i) = \frac{[\![|\mu_i| \le \min\{\frac{a_b a_v}{|\beta_i|}, a_w, a_v\}]\!]}{4a_b a_w a_v}\left(\min\{\frac{a_b}{|\beta_i|}, a_w\} - \frac{|\mu_i|}{a_v}\right)$$

*Proof.* Starting with Lemma 1,

$$f_{\beta,\mu}(\beta, \mu) = \int_{-a_w}^{a_w} f_B(\beta u) f_W(u) f_V(\mu/u) \, du$$

$$= \int_{-a_w}^{a_w} \frac{1}{2a_b}[\![-a_b \le \beta u \le a_b]\!] \frac{1}{2a_w}[\![-a_w \le u \le a_w]\!] \frac{1}{2a_v}[\![-a_v \le \mu/u \le a_v]\!] \, du$$

$$= \int_{-a_w}^{a_w} \frac{1}{2a_b}[\![-a_b/|\beta| \le u \le a_b/|\beta|]\!] \frac{1}{2a_w}[\![-a_w \le u \le a_w]\!] \frac{1}{2a_v}[\![u \le -|\mu|/a_v \vee u \ge |\mu|/a_v]\!] \, du$$

$$= \int_{-a_w}^{a_w} \frac{1}{8a_b a_w a_v}[\![-\min\{a_b/|\beta|, a_w\} \le u \le -|\mu|/a_v \vee |\mu|/a_v \le u \le \min\{a_b/|\beta|, a_w\}]\!]$$
$$\times [\![|\mu| \le a_b a_v/|\beta|]\!] \, du$$

$$= \frac{[\![|\mu| \le a_b a_v/|\beta|]\!]}{8a_b a_w a_v} \int_{-a_w}^{a_w} [\![-\min\{a_b/|\beta|, a_w\} \le u \le -|\mu|/a_v \vee |\mu|/a_v \le u \le \min\{a_b/|\beta|, a_w\}]\!] \, du$$

$$= \frac{[\![|\mu| \le a_b a_v/|\beta|]\!]}{4a_b a_w a_v} \int_0^{a_w} [\![|\mu|/a_v \le u \le \min\{a_b/|\beta|, a_w\}]\!] \, du$$

$$= \frac{[\![|\mu| \le a_b a_v/|\beta|]\!]}{4a_b a_w a_v} \left(\min\{a_b/|\beta|, a_w\} - |\mu|/a_v\right) [\![-a_w a_v \le \mu \le a_w a_v]\!]$$

as desired. $\square$

**Corollary 1(b).** *Consider the same setting as Theorem 1. In the case of an independent Uniform initialization,*

$$p_\beta(\beta_i) = \frac{1}{4a_b a_w}\left(\min\left\{\frac{a_b}{|\beta_i|}, a_w\right\}\right)^2$$

$$p_\mu(\mu_i) = \frac{[\![-a_w a_v \le \mu_i \le a_w a_v]\!]}{2a_w a_v} \log \frac{a_w a_v}{|\mu_i|}$$

$$p_{\mu|\beta}(\mu_i|\beta_i) = Tri(\mu_i; a_v \min\{a_b/|\beta_i|, a_w\}) = \frac{[\![|\mu_i| \le a_v \min\{a_b/|\beta_i|, a_w\}]\!]}{a_v \min\{a_b/|\beta_i|, a_w\}}\left(1 - \frac{|\mu_i|}{a_v \min\{a_b/|\beta_i|, a_w\}}\right),$$

*where $Tri(\cdot; a)$ is the symmetric triangular distribution with base $[-a, a]$ and mode $0$.*

*Proof.* Beginning with the marginal of $\beta_i$,

$$f_\beta(\beta) = \int f_{\beta,\mu}(\beta, \mu) \, d\mu$$

$$= \int_{-\infty}^{\infty} \frac{[\![|\mu| \le a_b a_v/|\beta|]\!]}{4a_b a_w a_v}\left(\min\{a_b/|\beta|, a_w\} - |\mu|/a_v\right) [\![-a_w a_v \le \mu \le a_w a_v]\!] \, d\mu$$

$$= \int_{-a_w a_v}^{a_w a_v} \frac{[\![|\mu| \le a_b a_v/|\beta|]\!]}{4a_b a_w a_v}\left(\min\{a_b/|\beta|, a_w\} - |\mu|/a_v\right) \, d\mu$$

$$= \frac{1}{2a_b a_w a_v} \int_0^{a_w a_v} [\![\mu \le a_b a_v/|\beta|]\!]\left(\min\{a_b/|\beta|, a_w\} - \mu/a_v\right) \, d\mu$$

$$= \frac{1}{2a_b a_w a_v} \left( \int_0^{a_w a_v} [\![\mu \le a_b a_v / |\beta|]\!] \min\{a_b/|\beta|, a_w\} \, d\mu - \int_0^{a_w a_v} [\![\mu \le a_b a_v / |\beta|]\!] \mu / a_v \, d\mu \right)$$

$$= \frac{1}{2a_b a_w a_v} \left( \min\{a_b/|\beta|, a_w\} \int_0^{a_w a_v} [\![\mu \le a_b a_v / |\beta|]\!] \, d\mu - \frac{1}{a_v} \int_0^{a_w a_v} [\![\mu \le a_b a_v / |\beta|]\!] \mu \, d\mu \right)$$

$$= \frac{1}{2a_b a_w a_v} \left( \min\{a_b/|\beta|, a_w\} \min\{a_w a_v, a_b a_v / |\beta|\} - \frac{1}{a_v} \int_0^{\min\{a_w a_v, a_b a_v / |\beta|\}} \mu \, d\mu \right)$$

$$= \frac{1}{2a_b a_w a_v} \left( \min\{a_b/|\beta|, a_w\} \min\{a_w a_v, a_b a_v / |\beta|\} - \frac{1}{2a_v} \left( \min\{a_w a_v, a_b a_v / |\beta|\} \right)^2 \right)$$

$$= \frac{1}{2a_b a_w a_v} \left( a_v \left( \min\{a_b/|\beta|, a_w\} \right)^2 - \frac{1}{2a_v} \left( a_v \min\{a_w, a_b/|\beta|\} \right)^2 \right)$$

$$= \frac{1}{2a_b a_w a_v} \left( a_v \left( \min\{a_b/|\beta|, a_w\} \right)^2 - \frac{a_v}{2} \left( \min\{a_w, a_b/|\beta|\} \right)^2 \right)$$

$$= \frac{1}{4a_b a_w} \left( \min\{a_b/|\beta|, a_w\} \right)^2$$

as desired. Then,

$$f_\mu(\mu) = \int f_{\beta,\mu}(\beta, \mu) \, d\beta$$

$$= \int_{-\infty}^{\infty} \frac{[\![|\mu| \le a_b a_v / |\beta|]\!]}{4a_b a_w a_v} \left( \min\{a_b/|\beta|, a_w\} - |\mu|/a_v \right) [\![-a_w a_v \le \mu \le a_w a_v]\!] \, d\beta$$

$$= \frac{[\![-a_w a_v \le \mu \le a_w a_v]\!]}{4a_b a_w a_v} \int_{-\infty}^{\infty} [\![|\mu| \le a_b a_v / |\beta|]\!] \left( \min\{a_b/|\beta|, a_w\} - |\mu|/a_v \right) d\beta$$

$$= \frac{[\![-a_w a_v \le \mu \le a_w a_v]\!]}{4a_b a_w a_v} 2 \int_0^{\infty} [\![|\mu| \le a_b a_v / \beta]\!] \left( \min\{a_b/\beta, a_w\} - |\mu|/a_v \right) d\beta$$

$$= \frac{[\![-a_w a_v \le \mu \le a_w a_v]\!]}{4a_b a_w a_v} 2 \int_0^{\infty} [\![\beta \le a_b a_v / |\mu|]\!] \left( \min\{a_b/\beta, a_w\} - |\mu|/a_v \right) d\beta$$

$$= \frac{[\![-a_w a_v \le \mu \le a_w a_v]\!]}{4a_b a_w a_v} 2 \int_0^{a_b a_v / |\mu|} \min\{a_b/\beta, a_w\} - |\mu|/a_v \, d\beta$$

$$= \frac{[\![-a_w a_v \le \mu \le a_w a_v]\!]}{4a_b a_w a_v} 2 \left( \int_0^{\min\{a_b a_v / |\mu|, a_b/a_w\}} a_w - |\mu|/a_v \, d\beta \right.$$

$$\left. + \int_{\min\{a_b a_v / |\mu|, a_b/a_w\}}^{a_b a_v / |\mu|} a_b/\beta - |\mu|/a_v \, d\beta \right)$$

$$= \frac{[\![-a_w a_v \le \mu \le a_w a_v]\!]}{4a_b a_w a_v} 2 \left( \int_0^{a_b/a_w} a_w - |\mu|/a_v \, d\beta + \int_{a_b/a_w}^{a_b a_v / |\mu|} a_b/\beta - |\mu|/a_v \, d\beta \right)$$

$$= \frac{[\![-a_w a_v \le \mu \le a_w a_v]\!]}{4a_b a_w a_v} 2 \left( (a_b/a_w)(a_w - |\mu|/a_v) + \int_{a_b/a_w}^{a_b a_v / |\mu|} a_b/\beta \, d\beta - \int_{a_b/a_w}^{a_b a_v / |\mu|} |\mu|/a_v \, d\beta \right)$$

$$= \frac{[\![-a_w a_v \le \mu \le a_w a_v]\!]}{4a_b a_w a_v} 2 \left[ (a_b/a_w)(a_w - |\mu|/a_v) + \int_{a_b/a_w}^{a_b a_v / |\mu|} a_b/\beta \, d\beta \right.$$

$$\left. - (|\mu|/a_v)(a_b a_v / |\mu| - a_b/a_w) \right]$$

$$= \frac{[\![-a_w a_v \le \mu \le a_w a_v]\!]}{4a_b a_w a_v} 2 \left( (a_b/a_w)(a_w - |\mu|/a_v) + a_b \log \frac{a_b a_v / |\mu|}{a_b/a_w} - (|\mu|/a_v)(a_b a_v / |\mu| - a_b/a_w) \right)$$

$$= \frac{[\![-a_w a_v \le \mu \le a_w a_v]\!]}{4a_b a_w a_v} 2a_b \log \frac{a_b a_v / |\mu|}{a_b/a_w}$$

$$= \frac{[\![-a_w a_v \le \mu \le a_w a_v]\!]}{2a_w a_v} \log \frac{a_b a_v / |\mu|}{a_b/a_w}$$

$$= \frac{[\![-a_w a_v \le \mu \le a_w a_v]\!]}{2a_w a_v} \log \frac{a_w a_v}{|\mu|}$$

as desired. We can then use these densities to derive the conditional:

$$f_\mu(\mu_i|\beta_i) = \frac{\frac{[\![|\mu_i| \le a_b a_v/|\beta_i|]\!]}{4a_b a_w a_v} \left(\min\{a_b/|\beta_i|, a_w\} - |\mu_i|/a_v\right) [\![-a_w a_v \le \mu_i \le a_w a_v]\!]}{\frac{1}{4a_b a_w} \left(\min\{a_b/|\beta_i|, a_w\}\right)^2}$$

$$= \frac{\frac{[\![|\mu_i| \le a_b a_v/|\beta_i|]\!][\![-a_w a_v \le \mu_i \le a_w a_v]\!]}{a_v} \left(\min\{a_b/|\beta_i * |, a_w\} - |\mu_i|/a_v\right)}{\left(\min\{a_b/|\beta_i|, a_w\}\right)^2}$$

$$= \frac{[\![|\mu_i| \le a_v \min\{a_b/|\beta_i|, a_w\}]\!]}{a_v \min\{a_b/|\beta_i|, a_w\}} \left(1 - \frac{|\mu_i|}{a_v \min\{a_b/|\beta_i|, a_w\}}\right).$$

$\square$

**Remarks.** Note that the marginal distribution on $\mu_i$ is the distribution of a product of two independent random variables, and the marginal distribution on $\beta_i$ is the distribution of the ratio of two random variables. For the Gaussian case, the marginal distribution on $\mu_i$ is a symmetric distribution with variance $\sigma_v^2 \sigma_w^2$ and excess Kurtosis of 6. For the Uniform case, the marginal distribution of $\beta_i$ is a symmetric distribution with no finite higher moments. The marginal distribution of $\mu_i$ is a symmetric distribution with bounded support and variance $\frac{2a_w^3 a_v^3}{9}$ and excess Kurtosis of $\frac{81}{50 a_w a_v} - 3$. The conditional distribution of $\mu_i$ given $\beta_i$ is a symmetric distribution with bounded support and variance $\frac{(a_v \min\{a_b/|\beta_i|, a_w\})^2}{6}$ and excess Kurtosis of $-\frac{3}{5}$.

### C.5 ROUGHNESS OF RANDOM INITIALIZATION

**Theorem 2.** *Consider the initial roughness $\rho_0$ under a Gaussian initialization. In the He initialization, we have that the tail probability is given by*

$$\mathbb{P}[\rho_0 \mathbb{E}[\rho_0] \ge \lambda] \le \frac{1}{1 + \frac{\lambda^2 H}{128}},$$

*where $\mathbb{E}[\rho_0] = 2$. In the Glorot initialization, we have that the tail probability is given by*

$$\mathbb{P}[\rho_0 \mathbb{E}[\rho_0] \ge \lambda] \le \frac{1}{1 + \frac{\lambda^2 (H+1)^4}{128 H}},$$

*where $\mathbb{E}[\rho_0] = \frac{2H}{(H+1)^2} = O\left(\frac{1}{H}\right)$.*

*Proof.* Using the moments of the delta-slope distribution computed in Theorem 1, Corollary 1, and above, we can compute:

$$\mathbb{E}[\rho_0] = \sum_{i=1}^{H} \mathbb{E}[\mu_i^2] = \sum_{i=1}^{H} \text{Var}[\mu_i] + \mathbb{E}[\mu_i]^2 = H\sigma_v^2 \sigma_w^2$$

$$\text{Var}[\rho_0] = \sum_{i=1}^{H} \text{Var}[\mu_i^2] = H\,\text{Var}[\mu_i^2] = H(\mathbb{E}[\mu_i^4] - \mathbb{E}[\mu_i^2]^2)$$

$$= H(9(\sigma_v \sigma_w)^4 - \sigma_v^4 \sigma_w^4) = 8H\sigma_v^4 \sigma_w^4$$

Applying the two initializations, we have

$$\mathbb{E}[\rho_0^{\text{He}}] = H\frac{2}{H}\frac{2}{1} = 4$$

$$\text{Var}[\rho_0^{\text{He}}] = 8H\frac{4}{H^2}\frac{4}{1} = \frac{128}{H}$$

$$\mathbb{E}[\rho_0^{\text{Glorot}}] = H\frac{2}{H+1}\frac{2}{1+H} = \frac{4H}{(H+1)^2} = O\left(\frac{1}{H}\right)$$

$$\text{Var}[\rho_0^{\text{Glorot}}] = 8H\frac{4}{(H+1)^2}\frac{4}{(1+H)^2} = \frac{128H}{(H+1)^4} = O\left(\frac{1}{H^3}\right)$$

By applying Cantelli's theorem, we get the stated tail probabilities. $\square$

C.6 DYNAMICS IN FUNCTION SPACE (BREAKPOINTS AND DELTA-SLOPES)

**Theorem 5.** *For a one hidden layer univariate ReLU network trained with gradient descent with respect to the neural network parameters $\theta_{NN} = \{(w_i, b_i, v_i)\}_{i=1}^{H}$, the gradient flow dynamics of the function space parameters $\theta_{BDSO} = \{(\beta_i, \mu_i)\}_{i=1}^{H}$ are governed by the following laws:*

$$\frac{d\beta_i}{dt} = -\frac{\partial \ell(\theta_{NN})}{\partial \beta_i} = \frac{v_i(t)}{w_i(t)} [\underbrace{\langle \hat{\boldsymbol{\epsilon}}(t) \odot \mathbf{a}_i(t), \mathbf{1} \rangle}_{\text{net relevant residual}} + \beta_i(t) \underbrace{\langle \hat{\boldsymbol{\epsilon}}(t) \odot \mathbf{a}_i(t), \mathbf{x} \rangle}_{\text{correlation}}] \tag{4}$$

$$\frac{d\mu_i(t)}{dt} = -\frac{\partial \ell(\theta_{NN})}{\partial \mu_i} = -(v_i^2(t) + w_i^2(t)) \langle \hat{\boldsymbol{\epsilon}}(t) \odot \mathbf{a}_i(t), \mathbf{x} \rangle - w_i(t)b_i(t) \langle \hat{\boldsymbol{\epsilon}}(t) \odot \mathbf{a}_i(t), \mathbf{1} \rangle \tag{5}$$

*Proof.* Computing the time derivatives of the BDSO parameters and using the loss gradients of the loss with respect to the NN parameters gives us:

$$\frac{\partial \ell(\theta_{NN})}{\partial w_i} = v_i \langle \hat{\boldsymbol{\epsilon}} \odot \mathbf{a}_i, \mathbf{x} \rangle$$

$$\frac{\partial \ell(\theta_{NN})}{\partial v_i} = \langle \hat{\boldsymbol{\epsilon}}, \sigma(w_i \mathbf{x} + b_i \mathbf{1}) \rangle = \langle \hat{\boldsymbol{\epsilon}} \odot \mathbf{a}_i, w_i \mathbf{x} + b_i \mathbf{1} \rangle = w_i \langle \hat{\boldsymbol{\epsilon}} \odot \mathbf{a}_i, \mathbf{x} \rangle + b_i \langle \hat{\boldsymbol{\epsilon}} \odot \mathbf{a}_i, \mathbf{1} \rangle$$

$$\frac{\partial \ell(\theta_{NN})}{\partial b_i} = v_i \langle \hat{\boldsymbol{\epsilon}} \odot \mathbf{a}_i, \mathbf{1} \rangle$$

$$\frac{d\beta_i(t)}{dt} = \frac{d}{dt} \left( -\frac{b_i(t)}{w_i(t)} \right)$$

$$= -\frac{w_i(t)\frac{db_i(t)}{dt} - b_i(t)\frac{dw_i(t)}{dt}}{w_i(t)^2}$$

$$= -\frac{w_i(t)(-\frac{\partial \ell(\theta_{NN})}{\partial b_i(t)}) - b_i(t)(-\frac{\partial \ell(\theta_{NN})}{\partial w_i(t)})}{w_i(t)^2}$$

$$= \frac{w_i(t)\frac{\partial \ell(\theta_{NN})}{\partial b_i(t)} - b_i(t)\frac{\partial \ell(\theta_{NN})}{\partial w_i(t)}}{w_i(t)^2}$$

$$= \frac{w_i(t)v_i(t)\langle \hat{\boldsymbol{\epsilon}}(t) \odot \mathbf{a}_i(t), \mathbf{1} \rangle - b_i(t)v_i(t)\langle \hat{\boldsymbol{\epsilon}}(t) \odot \mathbf{a}_i(t), \mathbf{x} \rangle}{w_i(t)^2}$$

$$= \frac{v_i(t) \langle \hat{\boldsymbol{\epsilon}}(t) \odot \mathbf{a}_i(t), w_i(t)\mathbf{1} - b_i(t)\mathbf{x} \rangle}{w_i(t)^2}$$

$$= \frac{v_i(t)}{w_i(t)} \left\langle \hat{\boldsymbol{\epsilon}}(t) \odot \mathbf{a}_i(t), \mathbf{1} - \frac{b_i(t)}{w_i(t)}\mathbf{x} \right\rangle$$

$$= \frac{v_i(t)}{w_i(t)} \left\langle \underbrace{\hat{\boldsymbol{\epsilon}}(t) \odot \mathbf{a}_i(t)}_{\text{relevant residuals}}, \mathbf{1} + \beta_i(t)\mathbf{x} \right\rangle$$

$$= \frac{v_i(t)}{w_i(t)} [\underbrace{\langle \hat{\boldsymbol{\epsilon}}(t) \odot \mathbf{a}_i(t), \mathbf{1} \rangle}_{\text{net relevant residual}} + \beta_i(t) \underbrace{\langle \hat{\boldsymbol{\epsilon}}(t) \odot \mathbf{a}_i(t), \mathbf{x} \rangle}_{\text{correlation}}]$$

$$\frac{d\mu_i(t)}{dt} = \frac{d}{dt} w_i v_i$$

$$= \frac{dw_i}{dt} v_i + w_i \frac{dv_i}{dt}$$

$$= -\frac{\partial \ell(\theta_{NN})}{\partial w_i} v_i - w_i \frac{\partial \ell(\theta_{NN})}{\partial v_i}$$

$$= -v_i^2 \langle \hat{\boldsymbol{\epsilon}} \odot \mathbf{a}_i, \mathbf{x} \rangle - w_i^2 \langle \hat{\boldsymbol{\epsilon}} \odot \mathbf{a}_i, \mathbf{x} \rangle - w_i b_i \langle \hat{\boldsymbol{\epsilon}} \odot \mathbf{a}_i, \mathbf{1} \rangle$$

$$= -(v_i^2 + w_i^2) \langle \hat{\boldsymbol{\epsilon}} \odot \mathbf{a}_i, \mathbf{x} \rangle - w_i b_i \langle \hat{\boldsymbol{\epsilon}} \odot \mathbf{a}_i, \mathbf{1} \rangle$$

This completes the proof. □

## C.7 Loss Surface in Function Space

**Theorem 3.** *Suppose that for all $p \in [P]$, $\hat{f}(\cdot; \theta_{BDSO})|_{\pi_p}$ is an Ordinary Least Squares fit of the data in piece $p$. Then, $\theta_{BDSO}$ is a critical point of $\tilde{\ell}(\theta_{BDSO})$.*

*Proof.* If, for all $p$, $\hat{f}(\cdot; \theta_{BDSO})|_{\pi_p}$ is an OLS fit of the data $\pi_p$, then we must have $\langle \hat{\boldsymbol{\epsilon}}_{\pi_p}, \pi_p \rangle = 0$, where $\hat{\boldsymbol{\epsilon}}_{\pi_p}$ is the residual for $\pi_p$. Similarly, we must have that the net residual $\langle \hat{\boldsymbol{\epsilon}}_{\pi_p}, \mathbf{1} \rangle = 0$.

Next, consider, for any neuron $j$, the vector $\hat{\mathbf{a}}_j$. If $j$ is right-facing, $\hat{\mathbf{a}}_j = (0, \ldots, 0, 1, \ldots, 1)$, where the transition from 0s to 1s corresponds to the data index $n$ where $x_n > \beta_j$; if $j$ is left-facing, a 1-to-0 transition occurs at $n$. Thus, $\hat{\mathbf{a}}_j$ is constant for $n \in \pi_p$, as the boundaries of $\pi_p$ correspond to breakpoints $\beta_i$ and $\beta_{i+1}$. Noting that these inner products are just sums of products, we have that, for any neuron $j$, $\langle \hat{\boldsymbol{\epsilon}} \odot \hat{\mathbf{a}}_j, \mathbf{x} \rangle$ can be decomposed into a sum $\sum_{\pi_p} \langle \hat{\boldsymbol{\epsilon}}_{\pi_p}, \pi_p \rangle = 0$, where the sum is over the pieces on the active side of $j$. Similarly, $\langle \hat{\boldsymbol{\epsilon}} \odot \hat{\mathbf{a}}_j, \mathbf{1} \rangle = \sum_{\pi_p} \langle \hat{\boldsymbol{\epsilon}}_{\pi_p}, \mathbf{1} \rangle = 0$.

Applying Theorem 5, we see that $\frac{d\beta_j}{dt} = \frac{d\mu_j}{dt} = 0$ for all $j$, and so $\theta_{BDSO}$ is a critical point of $\tilde{\ell}(\theta_{BDSO})$. $\qquad \square$

