# OpenReview forum: "A Functional Characterization of Randomly Initialized Gradient Descent in Deep ReLU Networks"
_ICLR.cc/2020/Conference — Reject_

### Official Review · AnonReviewer3 · 2019-10-23
**Official Blind Review #3**

**Rating:** 3

**Review:**


This paper proposes a functional characterization to understand the empirical success of deep neural networks. In particular, this paper focuses on the case of deep fully connected univariate ReLU networks, and show that the parameters will result in a Continuous Piecewise Linear (CPWL) approximation to the target function. Moreover, the authors derive the induced distributions of the function space parameters and show that increasing width can reduce the roughness of the initial function.

Besides, this paper analyzes the loss surface in the function space and reveals some relationship between the critical points in the function space and original NN parameter space. Furthermore, a type of gradient descent dynamic in the function space has also been derived.

Many experiments have been conducted to reveal how the expressiveness and optimization performance varies with the neural network width and depth.

Overall, the functional characterization is interesting can potentially help explain the generalization/expressiveness of deep neural networks. However, this paper is not well written and organized, and there are some “??'s” appearing on page 4. The authors should pay more attention to improving the writing and organization of this paper.

Here are the detailed comments:

The statements of theorems are not clear. For example, Theorems 3 and 4 convey too much information, I believe the authors should simplify the statements to make them more concise.  Moreover, the purpose of these two lemmas is not clear. Do they imply something related to expressiveness?
There should be some discussion in the surrounding text of Theorem 5. Some notations are also missing. For example, what’s $\hat \epsilon(t)$?, what’s $a_i(t)$? Besides, what’s the purpose of Theorem 5. It seems that this Theorem is used in the main part of this paper.
One major drawback of this paper is that it only provides the theoretical analysis for two-layer networks, but the title and claimed contributions are related to deep ReLU networks. I wonder whether the theory can be extended to deep cases?
In Theorem 2, I think He initialization will give you $\sigma_w = \sqrt{2/(H+1)}$ and $\sigma_v = \sqrt{2/k}$, if the output dimension is k. In this way, I am curious whether Theorem 2 can still hold. Besides, what’s the meaning of the so-called “roughness“?
I don’t get the point in Implications of Corollary 1. For example, what do you mean “$f$ has significant curvature in the boundaries?”, why an initialization that allocates more breakpoints to the area where the curvature of $f$ lies can be faster to train? The authors should elaborate more on this.


After reading the authors' response.

Thanks for your response.  I still think the contribution of this paper is not enough as the theoretical analysis may not be able to be generalized to deep networks. Thus I would like to keep my score.

**Experience Assessment:**

I have published one or two papers in this area.

**Review Assessment: Checking Correctness Of Derivations And Theory:**

I carefully checked the derivations and theory.

**Review Assessment: Checking Correctness Of Experiments:**

I assessed the sensibility of the experiments.

**Review Assessment: Thoroughness In Paper Reading:**

I read the paper at least twice and used my best judgement in assessing the paper.

---

> ### Author Response · Authors · 2019-11-15
> **Response to Review #3**
>
>
> We thank the reviewer for their helpful comments about issues with organization and clarity, particularly of Theorems 2-4.
>
> We updated our introduction and abstract to make it clear that the majority of our theory applies only to shallow ReLU networks, while experiments confirm that the qualitative theoretical predictions continue to hold in deeper networks. We hope to extend our theory to deeper networks in the future, but that is outside the scope of this work.
>
> Breakpoints are needed to model underlying function curvature. Therefore, if the initial breakpoints are far from the regions of curvature, it may take significant time for GD to move them. Table 1 shows this quantitatively and the new experiment (see response to reviewer 1 for more details) supports this, showing experimentally that breakpoint densities are effectively attracted towards the underlying function curvature during training. (Note that technically breakpoints are directly attracted to residual correlations as shown Eqn (4))
>
> We would like to thank the reviewer for catching an issue in Theorem 2; we are now using the correct definition of the He initialization.

---

### Official Review · AnonReviewer1 · 2019-10-24
**Official Blind Review #1**

**Rating:** 6

**Review:**

This paper wants to answer the question what is the value of the neural network’s depth? The author targets at Deep ReLU Network and uses Continuous Piecewise Linear (CWPL) to analyze the network’s parameter distribution. The main contributions of this paper are as follows, (1) For common initializations, this paper proves a deeper model will lead to flatter approximations and better approximation over a broader range of inputs. (2) A deeper model performs better when one optimizes with (Gradient Descent) GD methods. (3) Flat Initialization in the overparameterized regime could explain generalization. They found that the value of depth in deep nets seems less about expressivity, but enable GD to find better solutions.

My decision is Weak Accept, considering the following aspects.
Positive points: (1) The theory seems solid, authors prove breakpoint and delta-slope distribution will influence by the depth of the network. (2) The conclusions of the paper are inspiring, e.g., depth makes it easier for GD to the optimizer.

Negative points: (1) For the experiment, the authors find breakpoints can’t migrate very far from their initial location. I hope the author could explain this phenomenon since it is very crucial to proving the importance of the breakpoint’s initial distribution. (2) Some formulas in the appendix parts are beyond the scope of the page.

Suggestions: I think that a figure that shows breakpoint and input data distribution together will be very interesting. I want to see the breakpoint’s distribution change as training and its relationship with input data distribution.


**Experience Assessment:**

I do not know much about this area.

**Review Assessment: Checking Correctness Of Derivations And Theory:**

I did not assess the derivations or theory.

**Review Assessment: Checking Correctness Of Experiments:**

I assessed the sensibility of the experiments.

**Review Assessment: Thoroughness In Paper Reading:**

I read the paper at least twice and used my best judgement in assessing the paper.

---

> ### Author Response · Authors · 2019-11-15
> **Response to Review #1**
>
>
> We thank the reviewer for their comments about the effects of depth on breakpoint mobility. We agree that this is an important point, and have run  new experiments for several target functions(with results and figures now in the appendix) in order to better demonstrate this phenomenon. As suggested by the reviewer, we show the final breakpoint density in solved solutions from a shallow or deep neural network, compared against the underlying curvature of the ground truth function, along with the initial breakpoint density. In all cases, we show that the breakpoint distribution changes (at least somewhat) to better align with the target’s underlying curvature: For example, for a sine target function breakpoint-curvature correlation (BCC) increased significantly by .4 over the course of training, whereas for a  cubic function it only increased by .01. However, this effect is significantly more pronounced in deep networks, with the BCC increasing by .67 in the sine, and 1.59 in the cubic. The main results were added to the appendix, with some discussion in the first experimental section.

---

### Official Review · AnonReviewer2 · 2019-10-24
**Official Blind Review #2**

**Rating:** 3

**Review:**

The paper studies a number of interesting phenomena in deep learning by characterizing the linear regions of fully connected ReLU networks. The advantage of FC ReLU networks is that they are piecewise linear, and so the overall function can be understood in terms of these linear regions. The paper characterizes both the break points (boundaries) and slopes of these linear regions, using them to shed light on loss surfaces, generalization, and training dynamics.

The paper has interesting analyses, but I think the main drawback is that the clarity and presentation could be improved.

In particular, while reading the paper, I found myself wanting:
- More discussion of connections to relevant work. There have been a few papers (e.g. https://arxiv.org/abs/1611.01491, https://papers.nips.cc/paper/5422-on-the-number-of-linear-regions-of-deep-neural-networks.pdf, http://proceedings.mlr.press/v80/serra18b/serra18b.pdf) that use linear regions to understand deep networks. Reading this paper, I do not get a good understanding of how the work presented here fits into this larger research context.
- More expository text for particular concepts. A number of results are presented, and it would be helpful to have brief high-level summaries of the main findings of each section after diving through technical details.
- A number of terms are used before they are defined. For example, "roughness" is used on pages 1 and 2 but not defined until page 3. I think adding a preliminaries section with central terms/definitions in the paper clearly laid out could help with the exposition.

Other comments
- Consider defining breakpoints and delta-slopes when they are introduced at the bottom of pg. 1.
- typo on pg. 1: by doing *so* using small widths

**Experience Assessment:**

I do not know much about this area.

**Review Assessment: Checking Correctness Of Derivations And Theory:**

I assessed the sensibility of the derivations and theory.

**Review Assessment: Checking Correctness Of Experiments:**

I assessed the sensibility of the experiments.

**Review Assessment: Thoroughness In Paper Reading:**

I read the paper at least twice and used my best judgement in assessing the paper.

---

> ### Author Response · Authors · 2019-11-15
> **Response to Review #2**
>
> We thank the reviewer for pointing out additional related work. In particular, the paper by Serra et al. also took a function space view, characterizing ReLU networks as (C)PWLs, however our work focuses on training dynamics and generalization rather than expressive power. We added a short section acknowledging these papers in our Related Work section.

---

### Author Response · Authors · 2019-11-15
**General Response to Reviewers**


First, we wanted to draw the attention of the reviewers to our main theoretical result -- the explanation of implicit generalization in Section 2.4. To our knowledge, this is the first parsimonious explanation that shows how implicit regularization emerges as a joint consequence of (i) very flat initialization + gradient descent dynamics, (ii) overparameterization and (iii) a function parameterization that exhibits nonlocal residual dependence (i.e. breakpoints care about residuals far away from them). We have revised the abstract, introduction, and main contributions to more clearly emphasize this result.

Second, we thank the reviewers for their helpful comments regarding organization and clarity. We have worked to address these issues by making the following changes:

Added a reference to our definition of breakpoints and delta-slopes when they are introduced before being defined.
Added high-level summaries to Sec. 2.2, describing key findings
Rewrote the section describing the implications of corollary 1
Added a small section better introducing smoothness and our roughness metric.
Reworked Theorem 2 to be clearer, fixed an incorrect assumption
Reworked the statement of Theorem 3
Reworked the discussion following Theorem 3
Rewrote Theorem 4 as a shorter theorem and a remark
Moved Theorem 5 to the appendix - it wasn’t directly relevant to nearby theoretical results.
Significantly reworked and cleaned up the appendix, particularly the theory section
Various changes to the experimental sections to make them clearer and more concise

---

### Decision · Program_Chairs · 2019-12-19

**Decision:**

Reject

**Comment:**

This article sets out to study the advantages of depth and overparametrization in neural networks from the perspective of function space, with results on univariate shallow fully connected ReLU networks and some experiments on deep networks.
The article presents results on the concentration /dispersion of the slope / break point distribution of the functions represented by shallow univariate ReLU networks for parameters from various distributions. The reviewers found that the article contains interesting analysis, but that the presentation could be improved. The revision clarified some aspects and included some experiments illustrating breakpoint distributions in relation to the curvature of some target functions. However, the reviewers did not find this convincing enough, pointing out that the analysis focuses on a very restrictive setting and that that presentation of the article still could be improved. The discussion of implicit regularisation in section 2.4 seems promising, but it would benefit from a clearer motivation, background, and discussion.